# MIAU: Membership Inference Attack Unlearning Score for Quantifying the Forgetting Quality of Unlearning Methods

## Abstract

Machine unlearning aims to adapt the model's internal representations as if the forget set was never part of training set. In this context, a central challenge lies in accurately evaluating whether forgetting has actually occurred. Membership Inference Attacks (MIAs) are commonly used for this purpose; however, existing approaches are limited, often relying on single comparison and lacking reference points such as baseline and retrained model performance. We propose the Membership Inference Attack Unlearning Score (MIAU), a systematic metric that quantifies how closely an unlearning method mirrors the behavior of a fully retrained model. MIAU evaluates the unlearned model by comparing how easily it can separate three different pairs of data: forgotten samples versus test samples, forgotten samples versus retained samples, and retained samples versus test samples. These comparisons are then normalized between the performance of the original model and fully retrained model, providing an interpretable and balanced score of unlearning quality. The MIAU is intended to be used as an offline auditing benchmark to select the most suitable unlearning method for a given model setup and application setting, so that once this choice is made, the method can be applied in practice without performing any additional retraining. Extensive experiments demonstrate that MIAU consistently distinguishes effective unlearning methods across various image classification benchmarks and model architectures. Further statistical tests and empirical evaluations on retrained models—trained on 25%, 50%, and 75% of the forget set—highlight inherent limitations of MIAs in capturing gradual forgetting, presenting need for complementary evaluation methods in unlearning assessment.

## 1 Introduction

Machine learning (ML) models have achieved remarkable success across diverse applications, fueled by increasingly large datasets and powerful computing (LeCun et al., 2015; Jordan & Mitchell, 2015). This growth, however, has brought increased concerns about user privacy and data governance. Regulations such as the *European Union's General Data Protection Regulation (GDPR), specifically Article 17*, known as the "Right to be Forgotten", give individuals the legal authority to request the erasure of their personal data from digital systems (European Union, 2016). In the ML context, this has led to the emergence of machine unlearning which aims to update the model so that it behaves as if the designated data were never part of the training set, thereby complying with deletion requests while preserving the model's utility (Guo et al., 2020; Bourtoule et al., 2021; Cao & Yang, 2015; Ginart et al., 2019). However, a central challenge lies in rigorously verifying whether the influence of the forgotten data has indeed been eliminated, both in terms of the model's predictions and internal representations (Golatkar et al., 2020).

To address this, privacy-driven evaluation through membership inference attacks (MIAs) has become a cornerstone in auditing unlearning (Shokri et al., 2017; Chen et al., 2021b). MIAs aim to determine whether a sample was in the training set based on the model's outputs, serving as an empirical proxy for residual memorization (Shokri et al., 2017). In the context of unlearning, a successful method should render forgotten samples indistinguishable from unseen test data under such attacks.

## 1.1 PROBLEM STATEMENT

Despite the widespread adoption of membership inference attacks (MIAs) for evaluating unlearning, current evaluation methods remain limited. Most prior works compute MIA performance on only a single subset comparison—*Forget vs Test*, *Retain vs Forget*, or *Retain vs Test* (Chen et al., 2021a; Kurmanji et al., 2023; Chundawat et al., 2023a; Graves et al., 2021). Each captures a distinct aspect of model behavior, but relying on one gives an incomplete and potentially misleading picture. The *Forget vs Test* comparison measures whether the forget set remains more distinguishable than unseen data, capturing residual overfitting. Low separability here does not imply successful forgetting if the model has lost generalization and predictions become less confident. The *Retain vs Forget* comparison evaluates whether the model still treats forgotten samples like retained training data (Chundawat et al., 2023b). Because both sets come from the training distribution, effective unlearning should make model behavior on the forget set diverge from that on the retain set. High separability therefore signals successful forgetting, as the model no longer treats the forget set as part of training. However, separability alone cannot distinguish targeted forgetting from broader shifts in model behavior, such as unintended changes in handling retained data. The *Retain vs Test* setup measures whether the model behaves consistently on retained training data and unseen test data, serving as a sanity check for generalization. Although it does not directly show that the forget set was removed, it is essential for ruling out trivial explanations like underfitting or global degradation that could mimic forgetting. Thus, *Retain vs Test* provides context for interpreting the other two comparisons.

Each configuration examines a necessary but not sufficient condition for verifying unlearning. Evaluating only one cannot reveal whether a change in MIA performance comes from targeted forgetting or unrelated model degradation. Only by jointly analyzing all three can one isolate forgetting-specific effects from confounders such as underfitting, representation collapse, or loss of utility. An effective evaluation must integrate these perspectives into a unified measure capturing both completeness and correctness of forgetting.

Existing MIA-based evaluations also often lack proper baselines or reference points, making privacy gains hard to interpret. Many studies omit membership inference results on baseline or retrained models (Graves et al., 2021; Jia et al., 2023; Li et al., 2024). The baseline model, trained on the full dataset, represents worst-case privacy leakage. The retrained model, trained from scratch without the forget set, represents best-case "complete forgetting." An unlearned model's MIA score without these two reference points cannot show how much forgetting has been achieved or how close the method is to the ideal.

## 1.2 PROPOSED SOLUTION

While most prior evaluations omit such baselines, a limited number of studies retrain a model without the forget set to obtain a gold-standard benchmark, to use this reference solely for empirical, non-quantitative comparison with their proposed methods (Foster et al., 2024). While this single retrained reference is valuable for research, repeating such full retraining during deployment contradicts the purpose of machine unlearning and imposes additional computational cost. Therefore, in deployment settings, it is essential to establish a reliable metric that enables us to more accurately evaluate the practical utility of existing unlearning methods for our use case. Prior studies indicate that the effectiveness of unlearning techniques can vary significantly across different tasks (Cheng & Amiri, 2024). Consequently, identifying which approach is most suitable for our deployment scenario is critical, particularly given that continuous model retraining is infeasible in practice. To address this challenge, we introduce the Membership Inference Attack Unlearning Score (MIAU) as *an offline auditing framework* rather than an operational component of the unlearning pipeline. The resulting score computed once for each model–dataset pair guides researchers and practitioners in selecting the most effective unlearning method for their specific context. Thus, *it facilitates the consistent application of the most suitable unlearning method for a given model–dataset context, while eliminating the need for additional retraining overhead during model deployment.*

In this context, we propose the Membership Inference Attack Unlearning Score (MIAU)—a metric that quantifies how much of the performance difference between the original baseline model and the fully retrained model is closed by an unlearning method. MIAU captures the degree to which the unlearned model approximates the ideal privacy behavior of a retrained model that has never seen the forget set. It combines three complementary MIA comparisons—*Forget vs Test*, *Retain vs*

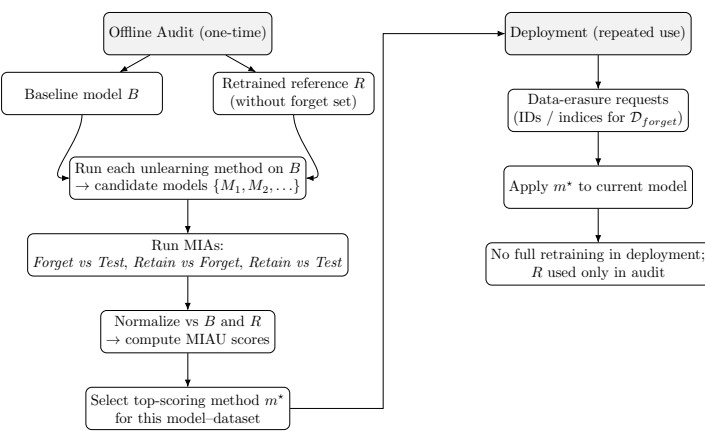

Figure 1: MIAU as a practical audit–deploy workflow. Left: a one-time offline audit selects the top-scoring method $m^\star$. Right: in deployment, data erasure requests are served by applying $m^\star$; no full retraining is performed.

*Forget*, and *Retain vs Test*—each measuring a distinct property of forgetting: residual memorization, removal effectiveness, and generalization stability, respectively. MIAU normalizes the unlearning method's performance between the baseline and retrain endpoints, producing a single interpretable score that reflects the completeness of forgetting. We evaluate MIAU on four standard image classification benchmarks—*MNIST* (LeCun et al., 1998), *CIFAR-10*, *CIFAR-20* (Krizhevsky & Hinton, 2009), and *MUCAC* (Choi & Na, 2023) —using three model architectures: *ResNet-18* (He et al., 2016), *All-CNN* (Springenberg et al., 2015), and *Vision Transformer* (Dosovitskiy et al., 2021). The evaluation includes four representative unlearning methods: *Fine-tune* (Bourtoule et al., 2021), *SSD* (Foster et al., 2024), *Amnesiac* (Graves et al., 2021), and *Teacher* (Chundawat et al., 2023b).

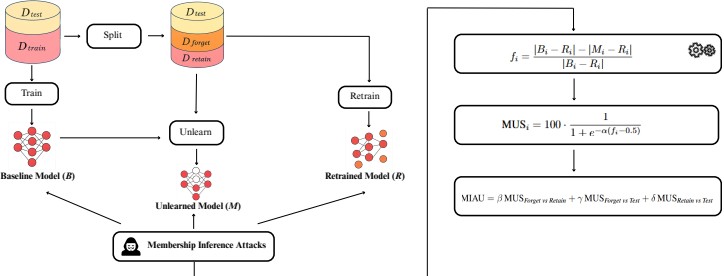

Figure 2: General pipeline of Membership Inference Attack Unlearning Score (MIAU) calculation.

Results demonstrate that MIAU provides a reliable and consistent measure of forgetting quality, distinguishing methods that closely approximate retraining from those that do not. Unlike raw MIA accuracy metrics, which are sensitive to attack strength, calibration shifts, or global degradation, MIAU provides an interpretable score between the *baseline* and *retrain* endpoints, enabling consistent comparisons across methods and datasets. To further assess its robustness, we evaluate MIAU under varying unlearning levels—removing 25%, 50%, and 75% of the forget set and assess improvements in score as more of the unlearning data is preserved. Statistical significance tests are further established via paired p-value tests comparing MIAU across methods.

## 2 RELATED WORK

Several machine unlearning studies assess forgetting by comparing model accuracy on the forget and retain subsets of the training data (Golatkar et al., 2020; 2021; Bourtoule et al., 2021). After unlearning, accuracy on the forget set is expected to drop slightly, ideally approaching test-level per-

formance, while retain-set accuracy should remain close to the original model or a retrained counterpart, indicating preserved utility on non-forgotten data. Additionally, test accuracy is typically used to ensure that overall generalization is not adversely affected by the unlearning process. However, relying on forget and retain accuracy alone introduces several limitations. A model can exhibit low forget-set accuracy by superficially suppressing predictions on the forget set, without eliminating the underlying learned representations (Golatkar et al., 2021; Nguyen et al., 2022). At the same time, a high retain accuracy does not guarantee that forgetting was targeted, as the model may have degraded uniformly or adapted in a way that preserves training performance without isolating the forgotten information. In contrast, MIAU evaluates all three membership-inference comparisons together, separating targeted forgetting from uniform degradation and clarifying whether accuracy changes reflect genuine unlearning or indiscriminate loss of predictive capacity.

To complement or replace these accuracy-based metrics, researchers have proposed dedicated unlearning evaluation scores. Zero-Retrain Forgetting (ZRF) (Chundawat et al., 2023b) score quantifies whether the model's predictions on the forget set become indistinguishable from those of a weak "incompetent" teacher. However, ZRF assumes that prediction randomness alone equates to forgetting, potentially overlooking residual information retained in the feature space. Meanwhile, the Normalized Machine Unlearning Score (NoMUS) (Choi & Na, 2023) balances forgetting and utility into a single normalized score. Despite its convenience, NoMUS relies on the particular formulation of the forgetting score and the fixed weight assigned to privacy versus utility, which may obscure whether high scores result from balanced forgetting and retention or from prioritizing one at the expense of the other. In contrast, MIAU combines the three MIA outcomes with explicitly adjustable coefficients, offering a transparent and theoretically grounded means to balance privacy protection for $\mathcal{D}_{\text{forget}}$ with predictive utility on $\mathcal{D}_{\text{retain}}$ and $\mathcal{D}_{\text{test}}$.

## 3 PRELIMINARIES

Let $\mathcal{D} = \{(x_i, y_i)\}_{i=1}^N$ be a supervised dataset with features $x_i \in \mathbb{R}^n$ and labels $y_i \in \{1, \ldots, K\}$. Machine unlearning aims to remove the influence of a subset $\mathcal{D}_{\text{forget}} \subset \mathcal{D}$ while retaining performance on the remaining data $\mathcal{D}_{\text{retain}} = \mathcal{D} \setminus \mathcal{D}_{\text{forget}}$. The model is modified so that its predictions and internal representations are indistinguishable from those of a model trained only on $\mathcal{D}_{\text{retain}}$.

We additionally define a disjoint test set $\mathcal{D}_{\text{test}}$ for generalization and privacy evaluation. The *baseline model* $\phi_{\theta_{\text{base}}}$ is trained on $\mathcal{D}$, and the *retrain model* $\phi_{\theta_{\text{retrain}}}$ is trained from scratch on $\mathcal{D}_{\text{retain}}$, representing ideal forgetting.

Unlearning quality is measured with three Membership Inference Attack (MIA) tasks, each implemented by training a binary classifier on model outputs to distinguish samples from two subsets. The *Forget vs Test* setup checks whether forgotten samples remain identifiable; perfect forgetting yields random guessing. The *Retain vs Forget* setup assesses removal effectiveness by testing separability between retained and forgotten data. The *Retain vs Test* setup evaluates whether retained data and unseen test data elicit similar predictions, reflecting generalization.

## 4 PROPOSED METHOD

The Membership Inference Attack Unlearning Score (MIAU) quantifies the extent to which an unlearning method approximates the privacy behavior of an ideal retrained model. It operates by comparing the outputs of the unlearned model to both the baseline and retrained models across multiple membership inference attack (MIA) tasks. Each task captures a distinct aspect of forgetting: residual memorization, separation between forgotten and retained data, and consistency on unseen samples. MIAU aggregates these evaluations into a single bounded score, enabling standardized assessment of unlearning effectiveness across different settings.

### 4.1 GAP CLOSURE FRACTION

Let the objective be to quantify the effectiveness of an unlearning method in approximating the privacy behavior of a model trained from scratch without the forgotten data. We consider three models evaluated on a given membership inference attack (MIA) task $i$: the *Baseline* model, trained on the full dataset; the *Retrain* model, trained with the forget set excluded from the beginning;

and the *Unlearning* model, which applies a forgetting algorithm to the baseline. Denote by $B_i$, $R_i$, and $M_i$ the membership-inference accuracies measured on task $i$ for the Baseline, Retrain, and Unlearning models, respectively.

To measure how much of the privacy gap between the Baseline and Retrain models has been closed by the unlearning method, we define the gap closure fraction as Eq. equation 1:

$$f_i = \frac{|B_i - R_i| - |M_i - R_i|}{|B_i - R_i|} \tag{1}$$

The quantity $f_i$ represents the relative reduction in distance to the retrain reference point. When $|B_i - R_i| = 0$, we directly set $f_i = 0$ to avoid division by zero. A value of $f_i = 1$ implies perfect alignment with the retrain model ($M_i = R_i$), indicating ideal forgetting. When $f_i = 0$, the unlearning method does not reduce the privacy gap relative to the baseline, and $f_i < 0$ indicates that the unlearning method increases the divergence from the retrain behavior.

## 4.2 MATCHING UNLEARNING SCORE ON METRIC $i$

While $f_i$ provides a normalized measure of forgetting effectiveness, it is unbounded and may be sensitive to outliers. To obtain a bounded and smooth score in the range $(0, 100)$, we apply a logistic transformation to $f_i$ and define the Matching Unlearning Score (MUS) as Eq. equation 2:

$$\text{MUS}_i = 100 \cdot \frac{1}{1 + e^{-\alpha(f_i - 0.5)}} \tag{2}$$

This formulation ensures several desirable properties. First, the score remains bounded in the open interval $(0, 100)$ without requiring manual clipping. Second, it centers the neutral reference point at $f_i = 0.5$, assigning a score of $\text{MUS}_i = 50$ to methods that close half the gap to retraining. Third, the parameter $\alpha$ controls the sensitivity of the transformation: larger values of $\alpha$ produce a steeper transition around the midpoint, amplifying differences in intermediate performance. To ensure that the $\text{MUS}_i$ score on *baseline* is near 0, and $\text{MUS}_i$ score on *retrain* is near 100, the $\alpha$ value for our setup was calculated to be 13.8. The detailed derivation of the $\alpha = 13.8$ calibration, which ensures that the $\text{MUS}_i$ score approaches 0 for the *baseline* and 100 for the *retrain* setting, is provided in Appendix A.1.

The logistic transformation is chosen because it maps the unbounded $f_i$ into a stable, interpretable percentage scale (0–100) while preserving the relative ordering of methods, enabling direct comparison across metrics. From a mathematical standpoint, the transformation satisfies the following limits: $\text{MUS}_i \to 0$ as $f_i \to -\infty$, $\text{MUS}_i = 50$ when $f_i = 0.5$, and $\text{MUS}_i \to 100$ as $f_i \to 1$. The logistic function is continuous and differentiable, making it suitable for ranking, visualization, or integration into gradient-based optimization procedures such as hyperparameter tuning or automated model selection.

## 4.3 MEMBERSHIP INFERENCE ATTACK UNLEARNING SCORE (MIAU)

To produce a unified assessment of forgetting quality across multiple MIA tasks, we define the final unlearning score as the average of the individual MUS values computed for each MIA setup. Specifically, the Membership Inference Attack Unlearning Score (MIAU) is given as Eq. equation 3:

$$\text{MIAU} = \beta \, \text{MUS}_{Forget\ vs\ Retain} + \gamma \, \text{MUS}_{Forget\ vs\ Test} + \delta \, \text{MUS}_{Retain\ vs\ Test} \tag{3}$$

The three MIA tasks correspond to the following subset pairs: $\mathcal{D}_{\text{forget}}$ vs. $\mathcal{D}_{\text{retain}}$ (*Forget vs Retain*), $\mathcal{D}_{\text{forget}}$ vs. $\mathcal{D}_{\text{test}}$ (*Forget vs Test*), and $\mathcal{D}_{\text{retain}}$ vs. $\mathcal{D}_{\text{test}}$ (*Retain vs Test*). Each task captures a distinct dimension of inference risk: the first reflects removal effectiveness, the second measures residual memorization, and the third assesses generalization consistency. The coefficients $\beta$, $\gamma$, and $\delta$ are non-negative weights that represent the relative importance assigned to each MIA direction and satisfy the constraints $\beta + \gamma + \delta = 1$ and $0 \leq \beta, \gamma, \delta \leq 1$. Depending on the desired emphasis, these weights can be chosen differently; however, for our experiments we set them equal ($\beta = \gamma = \delta = \frac{1}{3}$) to provide a balanced evaluation of unlearning performance, ensuring that no single MIA direction dominates the assessment.

## 5 EXPERIMENTAL SETUP

**Datasets and base models.** We evaluate the performance of our proposed MIAU score on four different datasets—*MNIST* (LeCun et al., 1998), *CIFAR-10*, its coarse-label variant *CIFAR-20* (Krizhevsky & Hinton, 2009), and the unlearning-specific face-attribute dataset *MUCAC* (Choi & Na, 2023). MNIST provides $28\times28$ grayscale digits (10 classes), CIFAR-10/20 contain $32\times32$ natural images with 10 and 20 superclasses, and MUCAC offers $128\times128$ celebrity portraits for binary smiling-attribute prediction. Models include *ResNet-18* (He et al., 2016), *All-CNN* (Springenberg et al., 2015), and the *Vision Transformer (ViT)* (Dosovitskiy et al., 2021). Dataset details, data preprocessing, training conditions, and all hyperparameter settings appear in Appendix A.5.

**Data splits.** Each dataset is partitioned into three disjoint subsets: the forget set $\mathcal{D}_{\text{forget}}$, the retain set $\mathcal{D}_{\text{retain}} = \mathcal{D}_{\text{train}} \setminus \mathcal{D}_{\text{forget}}$, and the test set $\mathcal{D}_{\text{test}}$, which is held out for evaluation. The forget set consists of 10% of the training data, sampled uniformly from each class to preserve the original class distribution. To ensure statistical robustness, the splitting and evaluation are repeated across 10 random seeds. For additional non-random evaluations, we also perform full-class forgetting of the `electric_devices` class in the CIFAR-20 All-CNN setup and 10% subclass forgetting of the `veg` category.

**Unlearning methods.** We evaluate four representative unlearning methods: Fine-tune, SSD, Amnesiac, and Teacher. Fine-tune (Bourtoule et al., 2021) retrains the model on the retain set with partial weight updates, SSD (Foster et al., 2024) penalizes parameters most influenced by the forget set, Amnesiac (Graves et al., 2021) reverses their gradient contributions, and Teacher (Chundawat et al., 2023b) distills knowledge from a full-data teacher into a student trained only on the retain set. All unlearning hyperparameters are detailed in Appendix A.5.4.

**Attack training protocol.** To quantify residual memorization and forgetting, we employ membership inference attacks using the model's softmax output distributions as input features. For each pair of data subsets involved in a given MIA task, a binary logistic regression classifier is trained to discriminate between them, similar to the setup offered by (Chundawat et al., 2023a). The training set for the attack model consists of 80% of the available logits, while 20% is held out for evaluation. Both entropy and maximum class confidence are implicitly captured in the softmax vectors, serving as indicators of memorization and decision margin. To avoid sampling bias during training, all subset pairs used in MIA tasks are size-matched by uniformly subsampling the larger set to match the cardinality of the smaller one. We further report the membership-inference accuracies obtained on the attack model's test split. Attack classifiers are trained independently for each of the three MIA setups: *Forget vs Test* ($\mathcal{D}_{\text{forget}}$ vs. $\mathcal{D}_{\text{test}}$), *Retain vs Forget* ($\mathcal{D}_{\text{retain}}$ vs. $\mathcal{D}_{\text{forget}}$), and *Retain vs Forget* ($\mathcal{D}_{\text{retain}}$ vs. $\mathcal{D}_{\text{forget}}$).

In addition to the softmax-based attack, we also evaluate a saliency-map–driven MIA. For this variant, we compute input-gradient saliency maps of the target model and XGBoost classifier to distinguish the saliency distributions of member and non-member samples, inspired by the attack setup in (Huang et al., 2024).

### 5.1 EVALUATION METRICS

**Gradual unlearning.** A desirable property of any unlearning evaluation metric is consistency under progressive removal of the forgotten data. That is, as a larger portion of the forget set is preserved in retraining, the unlearning score should increase accordingly. To verify that MIAU exhibits this behavior, we construct partial retraining baselines that simulate intermediate levels of forgetting.

Let $\mathcal{D}_{\text{forget}}^{(p)} \subset \mathcal{D}_{\text{forget}}$ denote a subset comprising a proportion $p \in \{0.25, 0.50, 0.75\}$ of the original forget set. The remaining portion $(1-p)$ is excluded from the unlearning procedure. The corresponding partial retrain set is given by $\mathcal{D}_{\text{retain}}^{(p)} = \mathcal{D}_{\text{retain}} \cup \mathcal{D}_{\text{forget}}^{(1-p)}$, and the model $\phi_{\theta_{\text{retrain-}p}}$ is trained from scratch on this subset.

These models represent intermediate stages of unlearning and serve as graded reference points between the baseline model (trained on $\mathcal{D}_{\text{retain}} \cup \mathcal{D}_{\text{forget}}$) and the full retrain model (trained solely

on $\mathcal{D}_{\text{retain}}$). We evaluate MIAU for Retrain 75%, Retrain 50%, and Retrain 25% to validate that the score increases as a larger proportion of the forget set is preserved. A consistent ordering of $\text{MIAU}_{25} < \text{MIAU}_{50} < \text{MIAU}_{75} < \text{MIAU}_{\text{full}}$ supports that MIAU faithfully captures the extent of forgetting.

**P-Value test.** To validate the consistency and discriminative capacity of MIAU under progressive unlearning, we perform statistical hypothesis testing across partial retrain levels. Specifically, we employ one-sided paired $t$-tests to evaluate whether the MIAU score at a higher unlearning level is significantly greater than at a lower level. For example, we test whether $\text{MIAU}_{50} > \text{MIAU}_{25}$, $\text{MIAU}_{75} > \text{MIAU}_{50}$, and $\text{MIAU}_{75} > \text{MIAU}_{25}$. All tests are conducted over multiple seeds, and we report the corresponding $p$-values to determine statistical significance at standard confidence levels.

## 6 DISCUSSION OF RESULTS

The results in Table 1 show that the proposed MIAU metric effectively quantifies unlearning performance by measuring how closely each method approximates the privacy behavior of a fully retrained model. For example, Amnesiac and Teacher achieve MIAU scores of 40.07% and 38.36%, respectively, meaning they close approximately 40% of the gap between the baseline model (MIAU $\approx$ 0.10%) and the ideal retraining model (MIAU $\approx$ 99.9%). In contrast, Finetune closes only 30.89%, and SSD just 8.55%, indicating less progress toward ideal forgetting. This design overcomes key limitations of individual MIA scores, which only reflect a single perspective and can be misleading without context. For instance, Amnesiac's *Forget vs Test* score is 56.58% and *Forget vs Retain* is 55.88%, but without knowing the corresponding baseline or retrain values, these numbers offer little interpretability. Similarly, accuracy-based metrics like *Forget Accuracy* (e.g., 83.99% for Amnesiac) and ZRF (95.95%) may not not distinguish between targeted forgetting and overall degradation. Each individual MIA configuration tests a necessary but not sufficient condition for successful unlearning. MIAU, on the other hand, provides a structured solution by integrating all three configurations and referencing the performance bounds, making it better suited for evaluating the completeness and correctness of forgetting. The remaining experimental results for all unlearning methods across additional datasets and configurations are provided in the Appendix A.7.

Table 1: Experiments on CIFAR-20 AllCNN

| Metric | Baseline | Amnesiac | Finetune | Teacher | SSD |
|---|---|---|---|---|---|
| Retain Accuracy | $90.2862 \pm 0.1332$ | $88.5177 \pm 0.1718$ | $86.8171 \pm 0.6593$ | $80.6776 \pm 0.6884$ | $90.3103 \pm 0.1159$ |
| Test Accuracy | $79.8926 \pm 0.0000$ | $76.7656 \pm 0.3392$ | $76.5400 \pm 0.4834$ | $69.2578 \pm 0.6123$ | $79.8936 \pm 0.0031$ |
| Forget Accuracy | $90.2968 \pm 0.4404$ | $83.9966 \pm 0.6690$ | $81.9508 \pm 0.7104$ | $78.8545 \pm 0.8649$ | $90.3999 \pm 0.3387$ |
| ZRF Score | $91.1786 \pm 0.0670$ | $95.9513 \pm 0.0945$ | $91.2336 \pm 0.3110$ | $96.8693 \pm 0.1198$ | $91.1546 \pm 0.0700$ |
| | | | | | |
| MIA (Forget vs Retain) | $50.2150 \pm 0.8686$ | $55.8800 \pm 1.3683$ | $51.8950 \pm 1.6075$ | $52.5300 \pm 0.8687$ | $50.6250 \pm 0.9044$ |
| MIA (Forget vs Test) | $49.6600 \pm 0.9703$ | $56.5800 \pm 0.8619$ | $51.8750 \pm 1.1660$ | $55.4050 \pm 0.8880$ | $49.5950 \pm 0.6950$ |
| MIA (Test vs Retain) | $49.7500 \pm 0.3450$ | $51.3775 \pm 0.7146$ | $50.3700 \pm 0.9752$ | $53.1900 \pm 0.5757$ | $49.6475 \pm 0.5362$ |
| MIA (Train vs Test) | $54.4750 \pm 0.0000$ | $55.5575 \pm 0.3060$ | $54.1125 \pm 0.7029$ | $53.8525 \pm 0.4610$ | $54.4750 \pm 0.0000$ |
| **MIAU** | $0.1007 \pm 0.0000$ | $40.0764 \pm 23.3662$ | $30.8884 \pm 15.2037$ | $38.3645 \pm 20.3545$ | $8.5516 \pm 13.4617$ |

The gradual unlearning results shown in Tables 2, and 3 confirm that MIAU scores increase monotonically with greater retraining data. For the three experiments, we observe that $\text{MIAU}_{25} < \text{MIAU}_{50} < \text{MIAU}_{75} < \text{MIAU}_{\text{full}}$ validating that MIAU reflects the expected partial unlearning behavior. This progressive rise aligns with the theoretical property of consistency under partial data removal, which is not inherently guaranteed by other metrics like forget accuracy or ZRF.

Despite this expected pattern in the tables, additional experiments uncover inherent limitations of MIAs. In particular, we can observe empirically, that the bar graph (Figure 3) shows that for several datasets, the expected progression $\text{MIAU}_{25} < \text{MIAU}_{50} < \text{MIAU}_{75} < \text{MIAU}_{\text{full}}$ does not hold consistently. For example, on `MNIST_AllCNN` and `CIFAR10_ResNet`, the increase in MIAU is not strictly observed across retraining levels. Statistically, the p-value heatmap (Figure 4) supports these irregularities: only a subset of datasets show meaningful p-values ($< 0.05$) for comparisons like $\text{MIAU}_{75} > \text{MIAU}_{25}$ or $\text{MIAU}_{75} > \text{MIAU}_{50}$. This suggests that individual MIA components (which our MIAU depends on) may still be unreliable in isolation. In particular, the bar graph

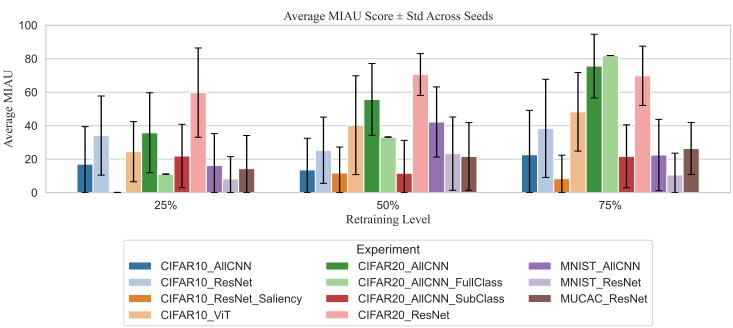

Figure 3: Average MIAU scores across 10 random seeds for each dataset at three retraining levels: 25%, 50%, and 75%.

Table 2: Gradual unlearning on CIFAR-10 ViT

| Metric | Retrain 25% | Retrain 50% | Retrain 75% | Retrain |
|---|---|---|---|---|
| Retain Accuracy | 98.7300 ± 0.4185 | 92.1279 ± 21.3489 | 89.9830 ± 28.0999 | 83.3823 ± 33.0112 |
| Test Accuracy | 97.1084 ± 0.3366 | 90.4990 ± 21.2237 | 88.5742 ± 27.5882 | 82.2012 ± 32.2264 |
| Forget Accuracy | 95.9102 ± 0.3428 | 89.7123 ± 20.3195 | 87.5362 ± 27.3287 | 81.1884 ± 31.9368 |
| ZRF Score | 77.2600 ± 0.6070 | 78.8569 ± 6.6877 | 76.7375 ± 0.6512 | 78.7411 ± 7.2286 |
| | | | | |
| MIA (Forget vs Retain) | 50.3000 ± 1.8720 | 53.1400 ± 3.8696 | 52.3667 ± 1.9312 | 52.5300 ± 2.2833 |
| MIA (Forget vs Test) | 52.2600 ± 1.5665 | 52.9000 ± 1.2944 | 52.4200 ± 1.4104 | 51.8600 ± 1.0723 |
| MIA (Test vs Retain) | 50.9500 ± 1.1756 | 51.2375 ± 0.9500 | 51.2750 ± 1.3260 | 51.1075 ± 1.0631 |
| MIA (Train vs Test) | 51.2500 ± 0.4518 | 51.4275 ± 0.7186 | 51.2325 ± 0.6836 | 51.0725 ± 0.6412 |
| **MIAU** | 24.4963 ± 17.9590 | 40.3098 ± 29.5193 | 48.2700 ± 23.4669 | 99.8993 ± 0.0000 |

(Figure 3) reveals substantial standard deviation across seeds for the same retraining level, indicating instability in per-run MIA behavior that may propagate into MIAU.

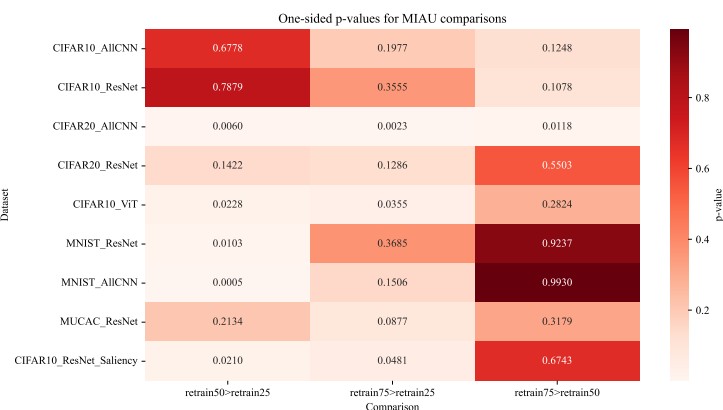

Figure 4: One-sided p-values from paired t-tests comparing MIAU scores between successive retraining levels across multiple datasets. Each cell reflects the statistical significance of whether the MIAU score from a higher retraining level is significantly greater than that of a lower one.

Further limitations may arise in scenarios where the model exhibits strong generalization across all data splits. As shown in Figure 5, MIA score distributions before and after unlearning often remain closely aligned, indicating minimal separability even when full retraining is performed. High variance in certain scores, especially under simple datasets like MNIST (see Figure 2), also implies that models with low memorization may naturally yield less distinct MIA signals. These observa-

Table 3: Gradual unlearning on MUCAC ResNet-18

| Metric | Retrain 25% | Retrain 50% | Retrain 75% | Retrain |
|---|---|---|---|---|
| Retain Accuracy | 93.4748 ± 2.1251 | 93.4056 ± 1.2176 | 92.8468 ± 2.3045 | 91.9656 ± 4.6271 |
| Test Accuracy | 93.4268 ± 1.8102 | 93.7485 ± 0.9387 | 93.1357 ± 1.9662 | 92.5286 ± 3.8284 |
| Forget Accuracy | 92.5586 ± 4.2547 | 92.9687 ± 0.8326 | 91.5788 ± 2.5153 | 90.9297 ± 4.5826 |
| ZRF Score | 76.1309 ± 4.2882 | 78.0748 ± 4.4559 | 78.4498 ± 6.1267 | 79.1186 ± 6.2524 |
| | | | | |
| MIA (Forget vs Retain) | 50.0000 ± 3.0162 | 49.4811 ± 3.2181 | 51.3249 ± 3.7769 | 50.4976 ± 1.9714 |
| MIA (Forget vs Test) | 53.5849 ± 4.1433 | 54.3396 ± 1.7338 | 55.7098 ± 2.1354 | 53.8863 ± 2.1638 |
| MIA (Test vs Retain) | 51.5254 ± 1.5284 | 51.9855 ± 1.1699 | 52.8208 ± 1.8111 | 52.3729 ± 1.0212 |
| MIA (Train vs Test) | 53.7772 ± 1.4660 | 52.7240 ± 0.8974 | 52.7119 ± 0.6782 | 52.7240 ± 1.5242 |
| **MIAU** | 14.3358 ± 19.8046 | 21.6582 ± 20.1905 | 26.3585 ± 15.5590 | 99.8993 ± 0.0000 |

tions suggest that in well-generalized regimes, MIAs may become less sensitive as an unlearning diagnostic, and supplementary indicators may be needed to confirm forgetting efficacy.

Detailed results of all experiment are outlined in Appendix A.7.

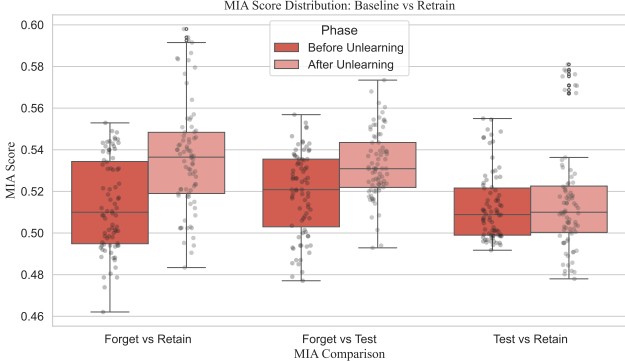

Figure 5: Comparison of MIA score distributions before and after unlearning. The figure illustrates the distributions of Membership Inference Attack (MIA) scores for three comparisons—*Forget vs Retain*, *Forget vs Test*, and *Test vs Retain*—across the *baseline* and *retrain* phases across all experiments.

## 7 CONCLUSION

This paper introduces the Membership Inference Attack Unlearning Score (MIAU), metric designed to provide more structured and interpretable performance assessment of machine unlearning methods. Unlike prior evaluation approaches that rely on a single MIA configuration or raw accuracy measures, MIAU integrates multiple attack comparisons—*Forget vs Test*, *Forget vs Retain*, and *Retain vs Test*. It further relates them to the baseline and fully retrained models to quantify the degree of gap closure. Through experiments across diverse datasets and model architectures, we illustrate that MIAU aligns with desirable properties for unlearning evaluation, including consistency under progressive removal and clear separation between effective and ineffective methods.

At the same time, our results highlight potential limitations in relying solely on MIAs for evaluation. In particular, we observe that individual MIA scores can be unstable across seeds and less informative for highly generalized models. These findings suggest that future work may investigate augmenting MIAU with models' internal behavioral indicators, such as latent space drift, neuron activation shifts, or feature attribution dynamics. As a result, these directions may help establish a more reliable understanding of the metric's sensitivity and consistency under varied unlearning regimes.

## LLM USAGE

Large Language Models (LLMs) were used only as a general-purpose writing assistant. They helped with grammar correction, phrasing, and minor style edits after the technical content, experiments, and analyses were completed by the authors. No part of the research ideation, methodology design or data analysis was generated by an LLM.

## REPRODUCIBILITY STATEMENT

To enable faithful reproduction of all results, we provide a complete specification of training and unlearning settings, implementation artifacts, and per-seed outputs. The full hyperparameter schedule is listed in Appendix A.5.4; dataset preprocessing and construction details (including any remapping and normalization) are in Appendix A.5.3; hardware and software environments are documented in Appendices A.4.1 and A.4.2; and training/validation loss traces used to monitor convergence are shown in Appendix A.5.5. The accompanying code and usage documentation (entry-point scripts, configuration files, and commands to regenerate tables and figures) are provided in the supplementary file `Code_Appendix.zip`. All per-seed experimental outputs—including metrics, logs, and CSVs of split indices—are contained in the supplementary file `Data_Appendix.zip`; using the fixed seeds and instructions included in these archives enables full reproduction of every figure and table in the paper.

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

# A APPENDIX

## A.1 CHOICE OF ALPHA PARAMETER

To ensure that the MUS score maps the gap closure fraction $f_i$ to values close to 0 for baseline models and close to 100 for fully retrained models, we select the steepness parameter $\alpha$ in the logistic transformation accordingly.

Recall that the MUS score is defined as:

$$\text{MUS}_i = 100 \cdot \frac{1}{1 + e^{-\alpha(f_i - 0.5)}} \tag{4}$$

We require the MUS score to satisfy:

$$\text{MUS}_i(f_i = 0) \approx 0.1 \tag{5}$$
$$\text{MUS}_i(f_i = 1) \approx 99.9 \tag{6}$$

Substituting $f_i = 0$ into Equation (1), we get:

$$100 \cdot \frac{1}{1 + e^{\alpha \cdot 0.5}} = 0.1 \tag{7}$$

$$\Rightarrow \quad \frac{1}{1 + e^{0.5\alpha}} = 0.001 \tag{8}$$

$$\Rightarrow \quad e^{0.5\alpha} = 999 \tag{9}$$

Solving for $\alpha$:

$$0.5\alpha = \ln(999) \tag{10}$$

$$\Rightarrow \quad \alpha = 2 \cdot \ln(999) \approx 2 \cdot 6.9068 = 13.8136 \tag{11}$$

Note that Equation (3) holds automatically by symmetry of the logistic function, given the same choice of $\alpha$ derived in Equation (2).

Therefore, setting $\alpha = 13.8$ ensures that the MUS score yields values close to 0.1 and 99.9 for the endpoints $f_i = 0$ and $f_i = 1$, respectively. This choice results in a sharp transition around $f_i = 0.5$ while maintaining bounded scores within $(0, 100)$, effectively amplifying the distinction between poorly and effectively unlearned models.

## A.2 PERFORMANCE OF STRONGER MIA ATTACKS

Table 4: Train vs. Test MIA Attack Performance on CIFAR-10 ResNet-18

| MIA Attack | Score (AUC) |
|---|---|
| LIRA | $44 \pm 5.8721$ |
| Shadow Model | $60 \pm 15.883$ |
| Quantile Regression | $50.9700 \pm 9.781$ |

## A.3 EFFECT SIZES AND CONFIDENCE INTERVALS

Table 5: Pairwise retraining–level comparisons across datasets

| Dataset | Comparison | n | Mean diff | 95% CI lower | 95% CI upper | Cohen's d | p-value |
|---|---|---|---|---|---|---|---|
| CIFAR10_AllCNN | retrain50 > retrain25 | 10 | -3.440051 | -19.742765 | 12.862664 | -0.150948 | 0.677751 |
| CIFAR10_AllCNN | retrain75 > retrain25 | 10 | 5.617811 | -8.623302 | 19.858924 | 0.282193 | 0.197713 |
| CIFAR10_AllCNN | retrain75 > retrain50 | 10 | 9.057862 | -7.586946 | 25.702669 | 0.389287 | 0.124755 |
| CIFAR10_ResNet | retrain50 > retrain25 | 10 | -8.750156 | -32.399339 | 14.899026 | -0.264681 | 0.787869 |
| CIFAR10_ResNet | retrain75 > retrain25 | 10 | 4.321670 | -21.245215 | 29.888556 | 0.120920 | 0.355526 |
| CIFAR10_ResNet | retrain75 > retrain50 | 10 | 13.071826 | -9.122867 | 35.266520 | 0.421318 | 0.107751 |
| CIFAR20_AllCNN | retrain50 > retrain25 | 10 | 19.914799 | 5.568661 | 34.260938 | 0.993033 | 0.005962 |
| CIFAR20_AllCNN | retrain75 > retrain25 | 10 | 39.839401 | 15.771734 | 63.907069 | 1.184136 | 0.002297 |
| CIFAR20_AllCNN | retrain75 > retrain50 | 10 | 19.924602 | 3.345701 | 36.503503 | 0.859719 | 0.011830 |
| CIFAR20_ResNet | retrain50 > retrain25 | 10 | 10.822943 | -10.688412 | 32.334298 | 0.359915 | 0.142223 |
| CIFAR20_ResNet | retrain75 > retrain25 | 10 | 10.051177 | -8.748205 | 28.850559 | 0.382469 | 0.128646 |
| CIFAR20_ResNet | retrain75 > retrain50 | 10 | -0.771766 | -14.208242 | 12.664710 | -0.041089 | 0.550262 |
| CIFAR10_ViT | retrain50 > retrain25 | 10 | 15.813495 | 0.375409 | 31.251582 | 0.732752 | 0.022849 |
| CIFAR10_ViT | retrain75 > retrain25 | 10 | 23.773717 | -2.499854 | 50.047288 | 0.647293 | 0.035483 |
| CIFAR10_ViT | retrain75 > retrain50 | 10 | 7.960222 | -22.164509 | 38.084952 | 0.189027 | 0.282369 |
| MNIST_ResNet | retrain50 > retrain25 | 10 | 15.072583 | 2.911955 | 27.233210 | 0.886655 | 0.010291 |
| MNIST_ResNet | retrain75 > retrain25 | 10 | 2.361378 | -13.059572 | 17.782327 | 0.109541 | 0.368502 |
| MNIST_ResNet | retrain75 > retrain50 | 10 | -12.711205 | -31.113758 | 5.691348 | -0.494119 | 0.923702 |
| MNIST_AllCNN | retrain50 > retrain25 | 10 | 26.016491 | 13.618136 | 38.414846 | 1.501092 | 0.000524 |
| MNIST_AllCNN | retrain75 > retrain25 | 10 | 6.293896 | -6.688195 | 19.275986 | 0.346815 | 0.150619 |
| MNIST_AllCNN | retrain75 > retrain50 | 10 | -19.722595 | -34.374532 | -5.070659 | -0.962924 | 0.993048 |
| MUCAC_ResNet | retrain50 > retrain25 | 10 | 7.322414 | -12.579052 | 27.223880 | 0.263204 | 0.213380 |
| MUCAC_ResNet | retrain75 > retrain25 | 10 | 12.022749 | -6.469253 | 30.514751 | 0.465096 | 0.087716 |
| MUCAC_ResNet | retrain75 > retrain50 | 10 | 4.700335 | -16.996604 | 26.397275 | 0.154972 | 0.317906 |
| CIFAR10_ResNet_Saliency | retrain50 > retrain25 | 10 | 11.629649 | 0.526751 | 22.732547 | 0.749295 | 0.020972 |
| CIFAR10_ResNet_Saliency | retrain75 > retrain25 | 10 | 8.229491 | -1.789463 | 18.248445 | 0.587589 | 0.048050 |
| CIFAR10_ResNet_Saliency | retrain75 > retrain50 | 10 | -3.400158 | -19.863704 | 13.063389 | -0.147740 | 0.674268 |

## A.4 COMPUTING INFRASTRUCTURE

### A.4.1 HARDWARE SPECIFICATIONS

Table 6: Hardware specifications of the Google Colab Pro+ environment with NVIDIA A100.

| Component | Specification |
|---|---|
| GPU | NVIDIA A100-SXM4-40GB, 40 GB HBM2 |
| CUDA Cores | 6,912 |
| Tensor Cores | 432 (3rd Generation) |
| GPU Memory Bandwidth | 1.6 TB/s |
| CUDA Version | 12.4 (runtime), 12.5 (nvcc compiler) |
| Driver Version | 550.54.15 |
| CPU | Intel(R) Xeon(R) @ 2.20GHz (12 vCPUs) |
| Threads per Core | 2 |
| Host RAM | 87.5 GB system memory |

### A.4.2 SOFTWARE SPECIFICATIONS

Table 7: Software packages and versions used in the Google Colab Pro+ A100 environment.

| Library / Component | Version |
|---|---|
| Operating System | Ubuntu 22.04.4 LTS (Jammy Jellyfish) |
| Python | 3.11.13 |
| PyTorch | 2.6.0+cu124 |
| Torchvision | 0.21.0+cu124 |
| CUDA Toolkit | 12.4 (linked), 12.5 (compiler nvcc) |
| NumPy | 2.0.2 |
| Pandas | 2.2.2 |
| Matplotlib | 3.10.0 |
| Pillow | 11.3.0 |
| scikit-learn | 1.6.1 |
| XGBoost | 3.0.2 |
| Transformers (HuggingFace) | 4.54.0 |
| TQDM | 4.67.1 |
| Seaborn | 0.13.2 |
| SciPy | 1.16.0 |
| Requests | 2.32.3 |

## A.5 TRAINING CONFIGURATIONS

### A.5.1 DATASETS

We evaluate on four image classification datasets (MNIST, Cifar-10, Cifar-20, MUCAC) with varying resolution, label structure, and domain characteristics. *MNIST* (LeCun et al., 1998) is a handwritten digit dataset with 10 classes (0–9), containing 60,000 training and 10,000 test grayscale images of size $28 \times 28$, uniformly distributed. *CIFAR-10* (Krizhevsky & Hinton, 2009) consists of natural $32 \times 32$ color images across 10 object categories, with 50,000 training and 10,000 test samples. *CIFAR-20* (Krizhevsky & Hinton, 2009) is a coarse-label variant of *CIFAR-100*, containing 20 superclasses (e.g., insects, vehicles) with 50,000 training and 10,000 test images, evenly split across classes. *MUCAC (Machine Unlearning for Celebrity Attribute Classifier)* (Choi & Na, 2023) is a binary attribute classification dataset derived from CelebA-HQ, used to evaluate unlearning on face attributes. For this study, we focus only on `Smiling` attribute as an independent binary classification task. The dataset contains of 10,548 training and 2,065 test samples, with 5284 negative and 4210 positive labels in each case.

### A.5.2 BASE MODELS

We evaluate unlearning performance across three representative architectures: ResNet-18, All-CNN, and Vision Transformer (ViT). *ResNet-18* (He et al., 2016) is a deep convolutional network utilizing residual connections to facilitate optimization and gradient flow. *All-CNN* (Springenberg et al., 2015) is a fully convolutional architecture with no max-pooling, preserving spatial locality and emphasizing robustness. *ViT* (Dosovitskiy et al., 2021) replaces convolutions with self-attention mechanisms, modeling global dependencies via patch embeddings and transformer layers.

### A.5.3 DATA PREPROCESSING

We adopt dataset-specific preprocessing strategies to enhance generalization performance and ensure consistency across training, unlearning, and evaluation phases. All image data are normalized to zero mean and unit variance using dataset-specific statistics.

For training from scratch on CIFAR-10 and CIFAR-20 datasets, we apply random cropping (with 4-pixel padding), random horizontal flipping, and random rotation of up to 15 degrees, followed by normalization using the dataset-specific mean $(0.5071, 0.4865, 0.4409)$ and standard deviation $(0.2673, 0.2564, 0.2762)$ values. For unlearning and test phases, no augmentation is applied; only normalization is used to maintain evaluation consistency. CIFAR-20 is constructed by remapping the

100 fine labels of CIFAR-100 into 20 coarse classes using a predefined superclass mapping derived from prior work.

MNIST images are first converted to three-channel grayscale format to align with the input expectations of RGB-based models. Training images are augmented with random rotation of up to 10 degrees. Test and unlearning images are not augmented. All images are normalized using a mean of 0.1307 and a standard deviation of 0.3081.

For MUCAC (CelebA-HQ) dataset, each face image is resized to $128 \times 128$ pixels. Training data are augmented with random horizontal flipping, affine transformation (shear angle of 10 degrees and scale factor between 0.8 and 1.2), and color jittering (brightness, contrast, and saturation set to 0.2). For unlearning and test data, only resizing and normalization are applied. The binary label is derived from the "smiling" attribute in the CelebA-HQ metadata. Images are split into training, forget, and test sets based on person identity ranges to enforce disjoint subsets.

### A.5.4 LIST OF HYPERPARAMETERS

Table 8: Summary of hyperparameters used for training and unlearning.

| Hyperparameter | Value |
|---|---|
| Batch size ($\mathcal{B}$) | 256 |
| Unlearning batch size | 128 |
| Initial learning rate ($\eta_0$) | 0.1 |
| Optimizer | SGD |
| Momentum | 0.9 |
| Weight decay | $5 \times 10^{-4}$ |
| Loss function | CrossEntropyLoss |
| Learning rate scheduler | MultiStepLR |
| Scheduler gamma ($\gamma$) | 0.2 |
| Warmup epochs | 1 |
| CIFAR-10 epochs | 20 |
| CIFAR-10 milestones | [8, 12, 16] |
| CIFAR-20 epochs | 40 |
| CIFAR-20 milestones | [15, 30, 35] |
| MNIST epochs | 5 |
| MNIST milestones | [2, 3, 4] |
| MUCAC epochs | 31 |
| MUCAC milestones | [10, 20] |
| CIFAR-10 (ViT) epochs | 8 |
| CIFAR-10 (ViT) milestones | [7] |
| Fine-tuning epochs | 5 |
| Fine-tuning learning rate | 0.02 |
| Amnesiac unlearning epochs | 3 |
| Amnesiac learning rate | 0.0001 |
| Dampening constant | 1 |
| Selection weighting | $10 \times$ model_size_scaler (default $= 10$) |
| Model size scaler | 1 |
| Device | GPU |

### A.5.5 TRAINING AND TEST LOSS ANALYSIS

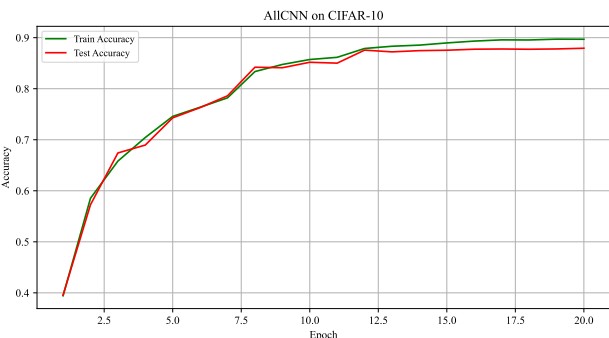

Figure 6: Training and validation loss and accuracy curves of the AllCNN model on the CIFAR-10 dataset.

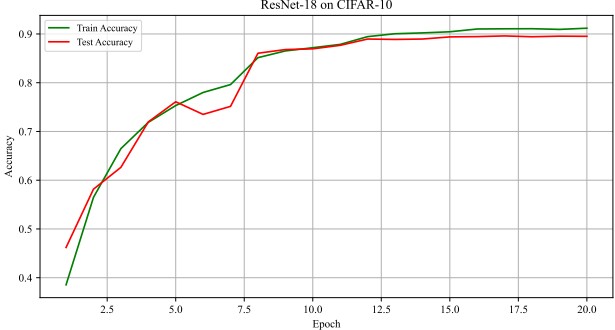

Figure 7: Training and validation loss and accuracy curves of the ResNet-18 model on the CIFAR-10 dataset.

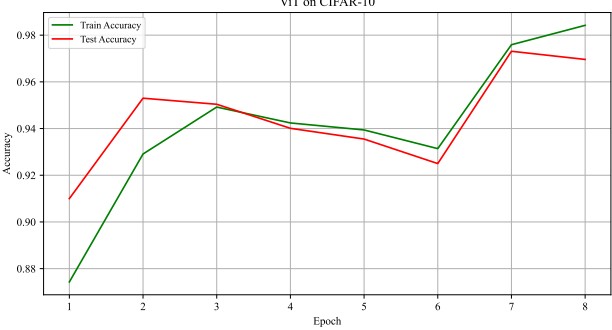

Figure 8: Training and validation loss and accuracy curves of the ViT model on the CIFAR-10 dataset.

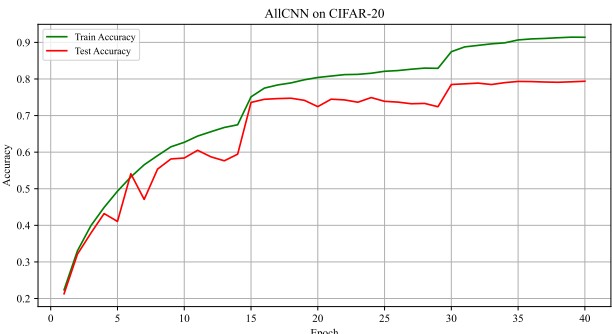

Figure 9: Training and validation loss and accuracy curves of the AllCNN model on the CIFAR-20 dataset.

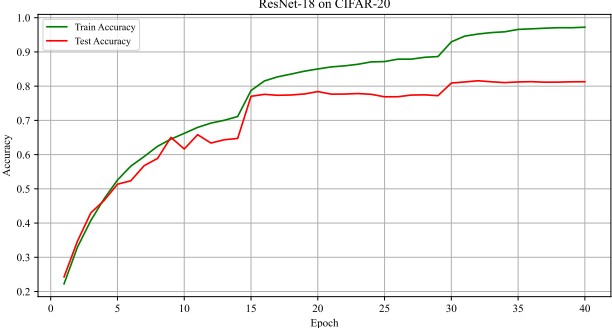

Figure 10: Training and validation loss and accuracy curves of the ResNet-18 model on the CIFAR-20 dataset.

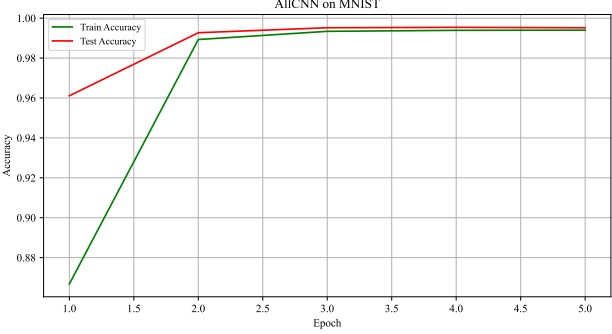

Figure 11: Training and validation loss and accuracy curves of the AllCNN model on the MNIST dataset.

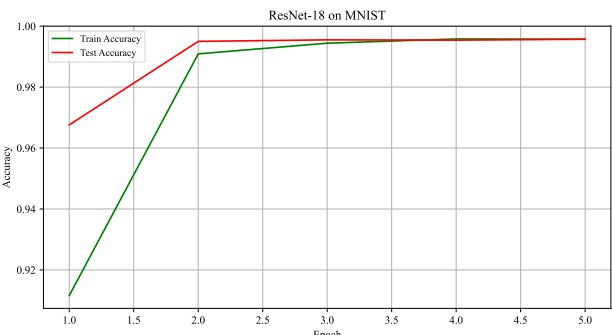

Figure 12: Training and validation loss and accuracy curves of the ResNet-18 model on the MNIST dataset.

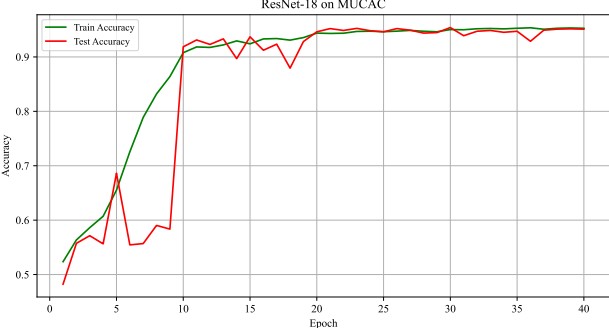

Figure 13: Training and validation loss and accuracy curves of the ResNet-18 model on the MUCAC dataset.

## A.6 RESULTS OF EXPERIMENTS ON NON-GENERALIZED MODELS

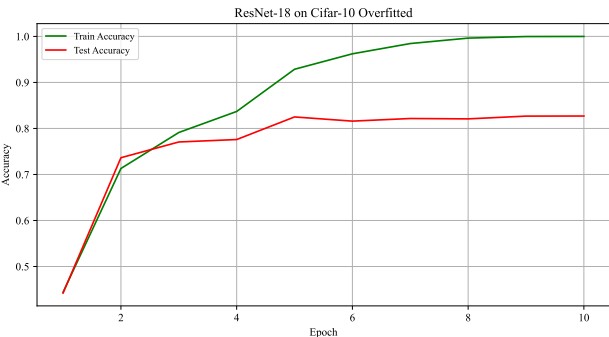

Figure 14: Training and validation loss and accuracy curves of the overfitted ResNet-18 model on the CIFAR-10 dataset. An underfitted model was obtained by taking the 1st epoch model, while overfitted obtained taking the 10th epoch model.

Table 9: Experiments on CIFAR10 underfitted ResNet-18

| Metric | Baseline | Amnesiac | Finetune | Teacher | SSD |
|---|---|---|---|---|---|
| Retain Accuracy | 44.8112 ± 0.0532 | 88.6696 ± 0.8549 | 90.2865 ± 2.0526 | 44.8467 ± 0.8808 | 44.8083 ± 0.0528 |
| Test Accuracy | 43.7793 ± 0.0000 | 76.9658 ± 0.8425 | 75.9062 ± 1.6650 | 43.8945 ± 0.7836 | 43.7793 ± 0.0000 |
| Forget Accuracy | 45.4627 ± 0.4842 | 71.4473 ± 1.1740 | 77.1919 ± 1.8680 | 45.6693 ± 1.0187 | 45.4247 ± 0.5211 |
| ZRF Score | 90.7447 ± 0.8915 | 95.6467 ± 0.5602 | 80.4635 ± 0.9641 | 98.3065 ± 0.1286 | 90.7469 ± 0.8908 |
| | | | | | |
| MIA (Forget vs Retain) | 56.2250 ± 0.8898 | 62.5450 ± 2.3834 | 55.2100 ± 4.2645 | 52.9350 ± 1.9764 | 56.4500 ± 1.0247 |
| MIA (Forget vs Test) | 50.0650 ± 0.6377 | 58.5450 ± 0.9576 | 49.2550 ± 0.7500 | 50.9700 ± 0.6638 | 49.9200 ± 0.6273 |
| MIA (Test vs Retain) | 60.0875 ± 0.2875 | 56.3450 ± 2.7112 | 58.0300 ± 4.0428 | 57.1175 ± 1.5528 | 60.0875 ± 0.2875 |
| MIA (Train vs Test) | 50.2000 ± 0.0000 | 53.5025 ± 0.4382 | 54.3250 ± 0.9622 | 50.5850 ± 0.4961 | 50.2000 ± 0.0000 |
| **MIAU** | 0.1007 ± 0.0000 | 6.9812 ± 13.4431 | 14.8824 ± 21.2491 | 31.2930 ± 22.8420 | 13.7048 ± 16.9481 |

Table 10: Gradual unlearning on CIFAR-10 underfitted ResNet-18

| Metric | Retrain 25% | Retrain 50% | Retrain 75% | Retrain |
|---|---|---|---|---|
| Retain Accuracy | 45.9210 ± 8.1250 | 48.7542 ± 8.5905 | 45.3409 ± 4.5958 | 47.8767 ± 6.4984 |
| Test Accuracy | 45.1240 ± 7.8977 | 47.6611 ± 8.5742 | 44.5781 ± 4.5092 | 46.8799 ± 6.5235 |
| Forget Accuracy | 45.2158 ± 8.2025 | 48.2742 ± 8.8719 | 45.0027 ± 4.0382 | 47.3834 ± 6.2705 |
| ZRF Score | 89.4216 ± 2.7347 | 89.5791 ± 4.4070 | 89.7330 ± 2.8897 | 90.6504 ± 1.5962 |
| | | | | |
| MIA (Forget vs Retain) | 60.3400 ± 5.4740 | 55.7600 ± 4.6058 | 54.4133 ± 3.7214 | 54.1800 ± 2.9321 |
| MIA (Forget vs Test) | 49.5200 ± 2.3117 | 50.5200 ± 1.7093 | 50.1200 ± 1.4208 | 50.1000 ± 0.4859 |
| MIA (Test vs Retain) | 59.9825 ± 4.5312 | 57.5825 ± 3.5780 | 58.4325 ± 3.9309 | 57.7300 ± 2.9314 |
| MIA (Train vs Test) | 49.6600 ± 0.5480 | 50.0700 ± 0.5820 | 50.0150 ± 0.5300 | 49.8100 ± 0.6517 |
| **MIAU** | 4.1438 ± 9.7553 | 21.9061 ± 19.9868 | 9.9082 ± 12.7532 | 96.5727 ± 10.5197 |

Table 11: Experiments on CIFAR-10 overfitted ResNet-18

| Metric | Baseline | Amnesiac | Finetune | Teacher | SSD |
|---|---|---|---|---|---|
| Retain Accuracy | 99.9947 ± 0.0019 | 95.6966 ± 0.8870 | 95.9652 ± 1.4784 | 95.6319 ± 1.2804 | 99.9947 ± 0.0019 |
| Test Accuracy | 82.8125 ± 0.0000 | 73.3311 ± 1.1328 | 78.1406 ± 1.2595 | 78.6924 ± 0.9585 | 82.8125 ± 0.0000 |
| Forget Accuracy | 99.9883 ± 0.0165 | 38.1286 ± 2.3546 | 84.2821 ± 1.9507 | 90.9142 ± 1.6170 | 99.9883 ± 0.0165 |
| ZRF Score | 75.2012 ± 0.3465 | 94.4499 ± 0.4395 | 73.7953 ± 1.2641 | 97.5783 ± 0.2209 | 75.2043 ± 0.3492 |
| | | | | | |
| MIA (Forget vs Retain) | 52.8300 ± 1.3756 | 68.5450 ± 7.5131 | 58.1250 ± 4.6867 | 55.5100 ± 3.2535 | 52.5500 ± 1.2530 |
| MIA (Forget vs Test) | 62.0400 ± 0.5739 | 63.3900 ± 1.2677 | 50.6100 ± 1.0011 | 52.6500 ± 0.5497 | 61.8750 ± 0.4152 |
| MIA (Test vs Retain) | 60.6925 ± 0.3283 | 57.9825 ± 5.9698 | 59.0950 ± 3.6548 | 53.6175 ± 2.8164 | 60.6925 ± 0.3283 |
| MIA (Train vs Test) | 62.6500 ± 0.0000 | 59.5725 ± 0.7982 | 58.0825 ± 0.8123 | 54.5425 ± 0.5683 | 62.6500 ± 0.0000 |
| **MIAU** | 0.1007 ± 0.0000 | 15.8030 ± 15.3861 | 56.2321 ± 15.9836 | 41.1594 ± 11.0159 | 0.1316 ± 0.0363 |

Table 12: Gradual unlearning on CIFAR-10 overfitted ResNet-18

| Metric | Retrain 25% | Retrain 50% | Retrain 75% | Retrain |
|---|---|---|---|---|
| Retain Accuracy | 99.9933 ± 0.0044 | 99.9931 ± 0.0037 | 99.9959 ± 0.0033 | 99.9956 ± 0.0033 |
| Test Accuracy | 82.6396 ± 0.4758 | 82.4072 ± 0.1979 | 82.2500 ± 0.3999 | 82.0928 ± 0.2645 |
| Forget Accuracy | 83.5998 ± 0.6553 | 83.3224 ± 0.6759 | 82.6646 ± 0.6620 | 82.6560 ± 0.9148 |
| ZRF Score | 77.2613 ± 0.7057 | 77.5656 ± 0.4064 | 77.8443 ± 0.4585 | 78.0136 ± 0.5318 |
| | | | | |
| MIA (Forget vs Retain) | 59.2200 ± 1.8890 | 60.5200 ± 1.2372 | 61.7200 ± 1.4476 | 61.4100 ± 1.2142 |
| MIA (Forget vs Test) | 49.8400 ± 1.8638 | 50.0900 ± 1.5624 | 49.4067 ± 1.6530 | 50.2250 ± 0.8193 |
| MIA (Test vs Retain) | 61.6125 ± 0.6727 | 61.8800 ± 0.5330 | 62.1150 ± 0.5953 | 62.0000 ± 0.6001 |
| MIA (Train vs Test) | 62.8850 ± 0.5599 | 63.0250 ± 0.5394 | 63.5125 ± 0.5562 | 62.7875 ± 0.3520 |
| **MIAU** | 69.2569 ± 14.8576 | 83.0650 ± 15.9770 | 81.5254 ± 20.3745 | 99.8993 ± 0.0000 |

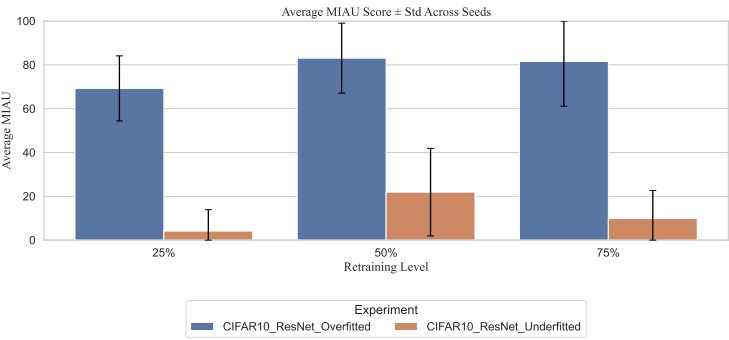

Figure 15: Average MIAU scores across 10 random seeds for underfitted and overfitted models at three retraining levels: 25%, 50%, and 75%.

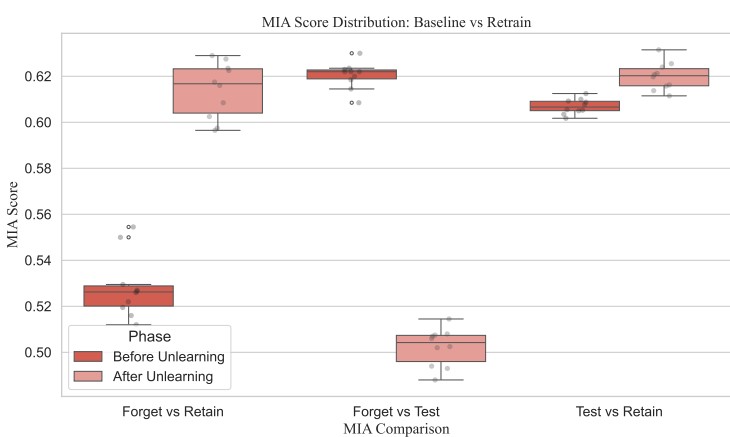

Figure 16: Comparison of MIA Score Distributions Before and After Unlearning for an overfitted model. The figure illustrates the distributions of Membership Inference Attack (MIA) scores for three comparisons—*Forget vs Retain*, *Forget vs Test*, and *Test vs Retain*—across the *baseline* and *retrain* phases across all experiments.

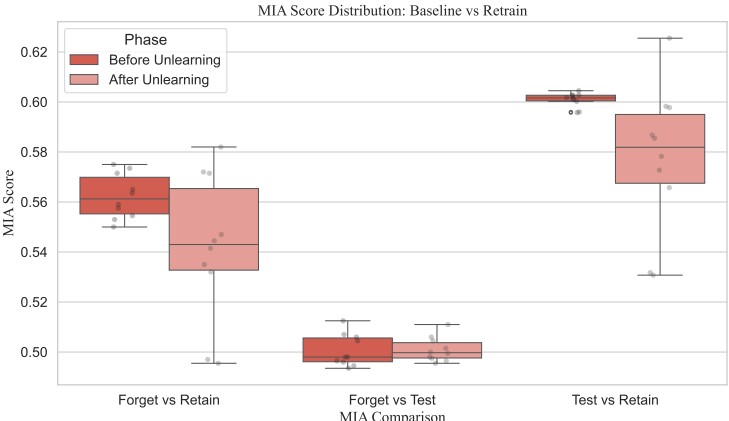

Figure 17: Comparison of MIA Score Distributions Before and After Unlearning for an underfitted model. The figure illustrates the distributions of Membership Inference Attack (MIA) scores for three comparisons—*Forget vs Retain*, *Forget vs Test*, and *Test vs Retain*—across the *baseline* and *retrain* phases across all experiments.

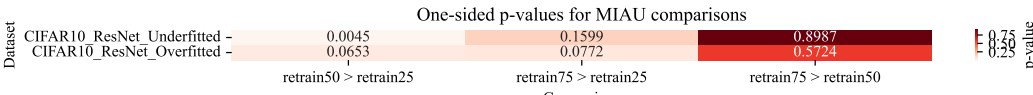

Figure 18: One-sided p-values from paired t-tests comparing MIAU scores between successive retraining levels across overfitted and underfitted models. Each cell reflects the statistical significance of whether the MIAU score from a higher retraining level is significantly greater than that of a lower one.

## A.7 REMAINING RESULTS OF EXPERIMENTS

Table 13: Experiments on CIFAR-10 AllCNN

| Metric | Baseline | Amnesiac | Finetune | Teacher | SSD |
|---|---|---|---|---|---|
| Retain Accuracy | 91.8457 ± 0.1097 | 90.5320 ± 0.2146 | 90.8454 ± 0.6374 | 84.1013 ± 0.4772 | 91.7852 ± 0.1036 |
| Test Accuracy | 89.5410 ± 0.0000 | 87.9102 ± 0.4012 | 88.2480 ± 0.5962 | 80.9824 ± 0.8379 | 89.5410 ± 0.0000 |
| Forget Accuracy | 91.8913 ± 0.4222 | 88.3699 ± 0.6600 | 88.6864 ± 0.7774 | 83.4465 ± 0.8203 | 91.9906 ± 0.2655 |
| ZRF Score | 84.2151 ± 0.2102 | 94.0819 ± 0.2160 | 83.3740 ± 0.5682 | 95.8960 ± 0.0819 | 84.2038 ± 0.2136 |
| | | | | | |
| MIA (Forget vs Retain) | 52.7100 ± 1.0775 | 52.3600 ± 3.7283 | 51.8750 ± 1.9332 | 53.7250 ± 1.9362 | 52.2300 ± 1.3290 |
| MIA (Forget vs Test) | 54.0900 ± 0.6927 | 58.1200 ± 0.9449 | 53.9700 ± 0.7262 | 58.5950 ± 0.8224 | 53.7800 ± 0.6808 |
| MIA (Test vs Retain) | 50.1400 ± 0.3710 | 53.8825 ± 1.6900 | 51.3000 ± 0.9587 | 55.2550 ± 1.3330 | 50.3025 ± 0.6309 |
| MIA (Train vs Test) | 51.8250 ± 0.0000 | 52.3575 ± 0.3939 | 51.7200 ± 0.6360 | 51.8200 ± 0.2986 | 51.8250 ± 0.0000 |
| **MIAU** | 0.1007 ± 0.0000 | 0.0263 ± 0.0796 | 12.9191 ± 14.5091 | 6.3614 ± 13.4248 | 19.4015 ± 17.2965 |

Table 14: Gradual unlearning on CIFAR-10 AllCNN

| Metric | Retrain 25% | Retrain 50% | Retrain 75% | Retrain |
|---|---|---|---|---|
| Retain Accuracy | 90.0336 ± 0.1156 | 89.8742 ± 0.1189 | 89.5516 ± 0.1528 | 89.4102 ± 0.1702 |
| Test Accuracy | 87.9609 ± 0.1674 | 87.8555 ± 0.2879 | 87.5352 ± 0.2244 | 87.3057 ± 0.2646 |
| Forget Accuracy | 86.7752 ± 0.9785 | 87.0890 ± 0.7315 | 86.2750 ± 0.5589 | 86.0535 ± 0.4969 |
| ZRF Score | 85.5322 ± 0.2350 | 85.6090 ± 0.1626 | 85.8212 ± 0.1960 | 85.8925 ± 0.2329 |
| | | | | |
| MIA (Forget vs Retain) | 51.2400 ± 2.6446 | 51.1200 ± 1.7390 | 50.6800 ± 1.7367 | 52.4300 ± 1.4074 |
| MIA (Forget vs Test) | 55.8200 ± 2.0099 | 55.8500 ± 1.1404 | 55.4733 ± 0.8500 | 54.8500 ± 0.8981 |
| MIA (Test vs Retain) | 50.3650 ± 0.5922 | 50.5100 ± 0.6868 | 50.1425 ± 0.6928 | 50.1000 ± 0.7773 |
| MIA (Train vs Test) | 51.6600 ± 0.2227 | 51.7225 ± 0.3754 | 51.4575 ± 0.3313 | 51.5100 ± 0.3526 |
| **MIAU** | 17.0318 ± 22.4070 | 13.5918 ± 18.9049 | 22.6496 ± 26.5159 | 99.8993 ± 0.0000 |

Table 15: Experiments on CIFAR-10 ResNet-18

| Metric | Baseline | Amnesiac | Finetune | Teacher | SSD |
|---|---|---|---|---|---|
| Retain Accuracy | 94.0087 ± 0.0768 | 91.4887 ± 0.3566 | 92.8625 ± 0.5377 | 92.5587 ± 0.1342 | 94.0446 ± 0.0929 |
| Test Accuracy | 91.5527 ± 0.0000 | 88.4434 ± 0.4419 | 89.9502 ± 0.5889 | 90.2285 ± 0.2665 | 91.5527 ± 0.0000 |
| Forget Accuracy | 94.0623 ± 0.1825 | 87.2577 ± 0.6693 | 90.4980 ± 0.8640 | 92.3504 ± 0.4102 | 94.0206 ± 0.3898 |
| ZRF Score | 82.4359 ± 0.3346 | 94.0667 ± 0.2474 | 82.3060 ± 0.4059 | 97.1861 ± 0.0486 | 82.4482 ± 0.3347 |
| | | | | | |
| MIA (Forget vs Retain) | 50.2400 ± 1.1279 | 55.8800 ± 4.3371 | 53.2550 ± 2.0196 | 51.9550 ± 1.0229 | 50.4250 ± 1.4111 |
| MIA (Forget vs Test) | 52.8450 ± 0.8217 | 52.5750 ± 1.6625 | 53.6600 ± 0.6847 | 56.6500 ± 1.4604 | 52.9300 ± 0.7196 |
| MIA (Test vs Retain) | 50.7300 ± 0.4362 | 54.2925 ± 3.9817 | 51.3375 ± 1.2198 | 53.7850 ± 1.8577 | 50.6225 ± 0.5369 |
| MIA (Train vs Test) | 52.6000 ± 0.0000 | 51.9925 ± 0.6261 | 52.6500 ± 0.3898 | 52.4550 ± 0.3427 | 52.6000 ± 0.0000 |
| **MIAU** | 0.1007 ± 0.0000 | 16.1329 ± 22.0528 | 32.2735 ± 29.7709 | 28.7886 ± 26.4404 | 12.7230 ± 15.8288 |

Table 16: Gradual unlearning on CIFAR-10 ResNet-18

| Metric | Retrain 25% | Retrain 50% | Retrain 75% | Retrain |
|---|---|---|---|---|
| Retain Accuracy | 91.2964 ± 0.1039 | 91.1475 ± 0.1444 | 90.9580 ± 0.1696 | 90.7916 ± 0.2118 |
| Test Accuracy | 89.4629 ± 0.1487 | 89.3711 ± 0.2428 | 88.9658 ± 0.2037 | 88.9658 ± 0.3181 |
| Forget Accuracy | 88.2205 ± 0.9465 | 87.8096 ± 0.4846 | 87.5664 ± 0.8027 | 87.3842 ± 0.6781 |
| ZRF Score | 83.0775 ± 0.4006 | 83.1148 ± 0.3020 | 83.2157 ± 0.3758 | 83.2750 ± 0.3216 |
| | | | | |
| MIA (Forget vs Retain) | 51.6600 ± 1.9845 | 51.8800 ± 1.5354 | 51.2667 ± 1.1226 | 52.7550 ± 1.2144 |
| MIA (Forget vs Test) | 55.7600 ± 1.9637 | 54.8900 ± 0.8346 | 53.8200 ± 0.8351 | 54.7000 ± 0.8100 |
| MIA (Test vs Retain) | 50.5350 ± 0.5424 | 50.3325 ± 0.6872 | 50.7650 ± 0.5525 | 50.6150 ± 0.5378 |
| MIA (Train vs Test) | 51.7550 ± 0.4923 | 52.0025 ± 0.2454 | 52.0975 ± 0.2454 | 51.8200 ± 0.4721 |
| **MIAU** | 34.0980 ± 23.6502 | 25.3478 ± 19.7844 | 38.4197 ± 29.3322 | 96.5727 ± 10.5197 |

Table 17: Gradual unlearning on CIFAR-20 AllCNN

| Metric | Retrain 25% | Retrain 50% | Retrain 75% | Retrain |
|---|---|---|---|---|
| Retain Accuracy | 92.0805 ± 0.1550 | 92.2980 ± 0.1561 | 92.1314 ± 0.1429 | 92.0245 ± 0.1341 |
| Test Accuracy | 78.7148 ± 0.3787 | 78.6826 ± 0.2449 | 78.4502 ± 0.3649 | 78.0762 ± 0.3496 |
| Forget Accuracy | 77.0443 ± 1.3364 | 77.4494 ± 0.6303 | 77.1019 ± 0.5649 | 76.6826 ± 0.3978 |
| ZRF Score | 91.3840 ± 0.1782 | 91.3772 ± 0.0858 | 91.4091 ± 0.1279 | 91.4487 ± 0.0747 |
| | | | | |
| MIA (Forget vs Retain) | 61.5962 ± 1.9613 | 53.4100 ± 1.3527 | 54.5066 ± 0.8483 | 54.7950 ± 1.2613 |
| MIA (Forget vs Test) | 53.9423 ± 1.5942 | 55.0400 ± 1.2295 | 54.0789 ± 0.7133 | 54.0200 ± 1.2406 |
| MIA (Test vs Retain) | 48.9350 ± 1.1443 | 49.5050 ± 1.3093 | 48.8900 ± 0.8253 | 49.1700 ± 1.1587 |
| MIA (Train vs Test) | 55.7850 ± 0.4802 | 56.4025 ± 0.2866 | 56.1475 ± 0.5581 | 55.6425 ± 0.3426 |
| **MIAU** | 35.7928 ± 23.8858 | 55.7076 ± 21.4602 | 75.6322 ± 19.0234 | 99.8993 ± 0.0000 |

Table 18: Experiments on CIFAR-20 ResNet-18

| Metric | Baseline | Amnesiac | Finetune | Teacher | SSD |
|---|---|---|---|---|---|
| Retain Accuracy | 95.2842 ± 0.0576 | 90.9696 ± 0.2383 | 92.5590 ± 0.6130 | 92.8336 ± 0.2617 | 95.2639 ± 0.0616 |
| Test Accuracy | 82.5977 ± 0.0000 | 76.6641 ± 0.3089 | 79.6943 ± 0.4911 | 80.4717 ± 0.3815 | 82.5977 ± 0.0000 |
| Forget Accuracy | 95.4085 ± 0.3380 | 78.9318 ± 0.3757 | 87.2881 ± 0.6136 | 91.3241 ± 0.5481 | 95.4053 ± 0.1522 |
| ZRF Score | 89.7542 ± 0.1137 | 95.5178 ± 0.1194 | 89.7126 ± 0.2155 | 98.0809 ± 0.0372 | 89.7701 ± 0.1413 |
| | | | | | |
| MIA (Forget vs Retain) | 49.4800 ± 0.6929 | 61.6450 ± 1.8385 | 52.9250 ± 1.2785 | 52.6700 ± 1.6371 | 49.5700 ± 0.7330 |
| MIA (Forget vs Test) | 53.8600 ± 0.7051 | 56.1200 ± 1.3931 | 50.2850 ± 1.1252 | 51.7850 ± 0.9548 | 53.7900 ± 0.7363 |
| MIA (Test vs Retain) | 54.7050 ± 0.5498 | 56.8850 ± 1.6878 | 54.2100 ± 1.0040 | 51.5200 ± 2.0472 | 54.7650 ± 0.7174 |
| MIA (Train vs Test) | 57.9000 ± 0.0000 | 56.6600 ± 0.7993 | 56.7075 ± 0.6874 | 55.2550 ± 0.5350 | 57.8950 ± 0.0158 |
| **MIAU** | 0.1007 ± 0.0000 | 42.7267 ± 10.9204 | 20.1822 ± 17.8346 | 20.3821 ± 17.9722 | 4.6739 ± 10.5222 |

Table 19: Gradual unlearning on CIFAR-20 ResNet-18

| Metric | Retrain 25% | Retrain 50% | Retrain 75% | Retrain |
|---|---|---|---|---|
| Retain Accuracy | 97.4987 ± 0.0893 | 97.5129 ± 0.0888 | 97.4814 ± 0.0790 | 97.3762 ± 0.1157 |
| Test Accuracy | 81.1885 ± 0.3370 | 81.1709 ± 0.2119 | 80.6758 ± 0.3016 | 80.5703 ± 0.2549 |
| Forget Accuracy | 79.5104 ± 1.2710 | 79.5967 ± 0.5677 | 79.0108 ± 0.6063 | 78.7410 ± 0.5979 |
| ZRF Score | 90.2851 ± 0.1700 | 90.2397 ± 0.1493 | 90.2327 ± 0.1394 | 90.2736 ± 0.1258 |
| | | | | |
| MIA (Forget vs Retain) | 62.7885 ± 1.8540 | 57.7300 ± 1.4453 | 58.5066 ± 1.1395 | 58.6900 ± 0.6867 |
| MIA (Forget vs Test) | 51.6346 ± 0.9077 | 53.0700 ± 0.8706 | 52.1776 ± 1.0492 | 52.3750 ± 1.1596 |
| MIA (Test vs Retain) | 56.8000 ± 0.6823 | 56.9975 ± 0.7163 | 57.1725 ± 0.5905 | 57.3350 ± 0.5027 |
| MIA (Train vs Test) | 60.0975 ± 0.5310 | 60.3700 ± 0.4489 | 60.0525 ± 0.4753 | 59.5050 ± 0.5350 |
| **MIAU** | 59.7610 ± 26.6740 | 70.5840 ± 12.5015 | 69.8122 ± 17.7220 | 99.8993 ± 0.0000 |

Table 20: Experiments on CIFAR-10 ViT

| Metric | Baseline | Amnesiac | Finetune | Teacher | SSD |
|---|---|---|---|---|---|
| Retain Accuracy | 98.7295 ± 0.0495 | 97.2495 ± 0.5614 | 98.8426 ± 0.3046 | 96.8133 ± 0.4246 | 98.7422 ± 0.0555 |
| Test Accuracy | 97.5684 ± 0.0000 | 95.7383 ± 0.3363 | 96.8184 ± 0.3160 | 96.1855 ± 0.3998 | 97.5723 ± 0.0179 |
| Forget Accuracy | 98.6807 ± 0.1439 | 91.7684 ± 0.7549 | 97.9156 ± 0.2678 | 95.7228 ± 0.5843 | 98.6376 ± 0.1340 |
| ZRF Score | 77.0079 ± 0.2782 | 94.6329 ± 0.2930 | 77.1782 ± 0.4557 | 96.8255 ± 0.3163 | 77.0329 ± 0.2635 |
| | | | | | |
| MIA (Forget vs Retain) | 50.0950 ± 0.7096 | 64.1800 ± 8.7837 | 51.8700 ± 1.2024 | 59.5600 ± 8.6347 | 49.9550 ± 0.8623 |
| MIA (Forget vs Test) | 49.5500 ± 0.5286 | 59.8750 ± 1.1275 | 49.5900 ± 0.8592 | 55.3350 ± 0.7885 | 49.1600 ± 0.6240 |
| MIA (Test vs Retain) | 49.7450 ± 0.2643 | 54.9600 ± 4.6806 | 51.8800 ± 0.9703 | 55.9925 ± 2.2616 | 49.8150 ± 0.3067 |
| MIA (Train vs Test) | 51.5000 ± 0.0000 | 51.4450 ± 0.7123 | 51.5275 ± 0.2982 | 51.6025 ± 0.7459 | 51.5025 ± 0.0606 |
| **MIAU** | 0.1007 ± 0.0000 | 3.7765 ± 10.4050 | 40.4777 ± 28.6503 | 12.8132 ± 15.7215 | 7.5180 ± 17.0937 |

Table 21: Experiments on MNIST ResNet-18

| Metric | Baseline | Amnesiac | Finetune | Teacher | SSD |
|---|---|---|---|---|---|
| Retain Accuracy | 99.5454 ± 0.0134 | 99.2985 ± 0.0867 | 99.7520 ± 0.0636 | 98.9609 ± 0.1205 | 99.5426 ± 0.0139 |
| Test Accuracy | 99.6484 ± 0.0000 | 99.3818 ± 0.0905 | 99.5312 ± 0.0385 | 99.2305 ± 0.0808 | 99.6484 ± 0.0000 |
| Forget Accuracy | 99.4852 ± 0.0952 | 98.8300 ± 0.1569 | 99.3034 ± 0.1016 | 98.8869 ± 0.1876 | 99.5140 ± 0.0787 |
| ZRF Score | 74.0607 ± 0.3455 | 94.4436 ± 0.3431 | 73.1390 ± 0.4803 | 97.2122 ± 0.1159 | 74.0628 ± 0.3678 |
| | | | | | |
| MIA (Forget vs Retain) | 53.8458 ± 0.4357 | 60.4708 ± 7.5796 | 52.8583 ± 1.6990 | 59.1333 ± 5.8952 | 53.8333 ± 0.5368 |
| MIA (Forget vs Test) | 51.0583 ± 1.5168 | 55.0167 ± 1.2387 | 50.4792 ± 1.6006 | 53.5250 ± 1.2663 | 51.5792 ± 1.1327 |
| MIA (Test vs Retain) | 51.3625 ± 0.3367 | 55.6150 ± 4.7899 | 51.9750 ± 1.2923 | 54.8375 ± 5.5330 | 51.3775 ± 0.2639 |
| MIA (Train vs Test) | 49.5000 ± 0.0000 | 49.8275 ± 0.7439 | 50.0525 ± 0.2425 | 50.2850 ± 0.4264 | 49.5000 ± 0.0000 |
| **MIAU** | 0.1007 ± 0.0000 | 5.4088 ± 11.3423 | 10.0907 ± 15.7429 | 4.5890 ± 10.2705 | 18.4736 ± 11.5964 |

Table 22: Gradual unlearning on MNIST ResNet-18

| Metric | Retrain 25% | Retrain 50% | Retrain 75% | Retrain |
|---|---|---|---|---|
| Retain Accuracy | 99.5755 ± 0.0262 | 99.5980 ± 0.0222 | 99.5696 ± 0.0326 | 99.5856 ± 0.0199 |
| Test Accuracy | 99.6152 ± 0.0167 | 99.6230 ± 0.0496 | 99.5938 ± 0.0463 | 99.6221 ± 0.0360 |
| Forget Accuracy | 99.2215 ± 0.2340 | 99.1044 ± 0.1805 | 99.2226 ± 0.1275 | 99.2399 ± 0.1455 |
| ZRF Score | 73.3765 ± 0.4144 | 73.5285 ± 0.4257 | 73.6286 ± 0.5342 | 73.8260 ± 0.4286 |
| | | | | |
| MIA (Forget vs Retain) | 53.4167 ± 1.7074 | 53.2250 ± 1.0252 | 52.1611 ± 0.6534 | 53.9333 ± 0.8318 |
| MIA (Forget vs Test) | 48.8333 ± 1.8257 | 50.8833 ± 0.5934 | 50.1056 ± 0.4825 | 52.0250 ± 0.3964 |
| MIA (Test vs Retain) | 51.6825 ± 0.4548 | 51.7400 ± 0.3784 | 51.9475 ± 0.2727 | 51.4500 ± 1.3956 |
| MIA (Train vs Test) | 49.6775 ± 0.2247 | 49.7275 ± 0.2314 | 49.6575 ± 0.1915 | 49.7075 ± 0.1799 |
| **MIAU** | 8.2382 ± 13.2938 | 23.3108 ± 21.8800 | 10.5996 ± 12.9944 | 99.8993 ± 0.0000 |

Table 23: Experiments on MNIST AllCNN

| Metric | Baseline | Amnesiac | Finetune | Teacher | SSD |
|---|---|---|---|---|---|
| Retain Accuracy | 99.4275 ± 0.0155 | 99.1839 ± 0.0782 | 99.3935 ± 0.1259 | 98.9995 ± 0.0863 | 99.4184 ± 0.0178 |
| Test Accuracy | 99.5312 ± 0.0000 | 99.3682 ± 0.0658 | 99.4189 ± 0.0792 | 99.2197 ± 0.0378 | 99.5312 ± 0.0000 |
| Forget Accuracy | 99.3776 ± 0.1103 | 98.8863 ± 0.1772 | 99.0297 ± 0.1797 | 98.9465 ± 0.1671 | 99.3438 ± 0.0835 |
| ZRF Score | 80.3363 ± 0.1385 | 94.7224 ± 0.2697 | 80.2219 ± 0.3555 | 96.9156 ± 0.1277 | 80.3471 ± 0.1326 |
| | | | | | |
| MIA (Forget vs Retain) | 54.3917 ± 0.4927 | 56.4500 ± 6.0605 | 55.6417 ± 5.0342 | 57.1333 ± 5.0901 | 54.4833 ± 0.8030 |
| MIA (Forget vs Test) | 51.9417 ± 1.5687 | 54.2500 ± 0.8382 | 52.1375 ± 0.4803 | 55.5542 ± 0.7126 | 52.3417 ± 0.3752 |
| MIA (Test vs Retain) | 52.6525 ± 0.3334 | 54.7600 ± 6.1997 | 53.2125 ± 3.0703 | 53.7450 ± 3.7670 | 52.6225 ± 0.2928 |
| MIA (Train vs Test) | 49.5250 ± 0.0000 | 50.1975 ± 0.4841 | 49.8625 ± 0.6183 | 49.8150 ± 0.5269 | 49.5250 ± 0.0000 |
| **MIAU** | 0.1007 ± 0.0000 | 7.8449 ± 13.1213 | 13.3123 ± 21.4386 | 6.8981 ± 20.8485 | 17.1314 ± 21.6850 |

Table 24: Gradual unlearning on MNIST AllCNN

| Metric | Retrain 25% | Retrain 50% | Retrain 75% | Retrain |
|---|---|---|---|---|
| Retain Accuracy | 99.4433 ± 0.0224 | 99.4407 ± 0.0199 | 99.4258 ± 0.0246 | 99.4264 ± 0.0151 |
| Test Accuracy | 99.5234 ± 0.0359 | 99.5312 ± 0.0305 | 99.5283 ± 0.0230 | 99.5156 ± 0.0377 |
| Forget Accuracy | 99.2454 ± 0.2434 | 99.1207 ± 0.1601 | 99.1803 ± 0.1209 | 99.1397 ± 0.1498 |
| ZRF Score | 80.0135 ± 0.2128 | 80.2209 ± 0.2102 | 80.3184 ± 0.1957 | 80.3267 ± 0.1638 |
| | | | | |
| MIA (Forget vs Retain) | 52.2500 ± 2.2295 | 52.7167 ± 1.1360 | 52.0333 ± 1.1568 | 53.5083 ± 0.7060 |
| MIA (Forget vs Test) | 51.1333 ± 2.4073 | 51.1417 ± 1.2373 | 50.2556 ± 0.6025 | 52.3375 ± 0.4947 |
| MIA (Test vs Retain) | 50.1225 ± 1.3757 | 51.1750 ± 0.9858 | 50.6325 ± 1.5158 | 50.2925 ± 1.6335 |
| MIA (Train vs Test) | 49.6225 ± 0.1913 | 49.6450 ± 0.2260 | 49.8000 ± 0.3173 | 49.9000 ± 0.3418 |
| **MIAU** | 16.2060 ± 19.0858 | 42.2225 ± 20.9768 | 22.4999 ± 21.3182 | 96.5727 ± 10.5197 |

Table 25: Experiments on MUCAC ResNet-18

| Metric | Baseline | Amnesiac | Finetune | Teacher | SSD |
|---|---|---|---|---|---|
| Retain Accuracy | 95.3532 ± 0.1437 | 91.9066 ± 2.5253 | 91.8050 ± 3.3542 | 87.4326 ± 2.9091 | 88.9334 ± 13.7919 |
| Test Accuracy | 95.8767 ± 0.0000 | 92.6241 ± 2.5054 | 92.2194 ± 2.6400 | 88.8514 ± 2.2473 | 90.1394 ± 13.2272 |
| Forget Accuracy | 95.3198 ± 0.7931 | 90.5474 ± 3.0677 | 91.1104 ± 3.6423 | 87.4781 ± 3.0574 | 88.9276 ± 13.9543 |
| ZRF Score | 72.6464 ± 1.5450 | 94.5932 ± 0.6404 | 76.5433 ± 5.2505 | 95.0513 ± 0.4298 | 76.9392 ± 6.3037 |
| | | | | | |
| MIA (Forget vs Retain) | 49.7393 ± 2.7892 | 50.1422 ± 2.8087 | 50.5687 ± 2.0634 | 50.8294 ± 1.1569 | 49.6445 ± 3.1640 |
| MIA (Forget vs Test) | 52.3697 ± 1.9668 | 53.4360 ± 3.1719 | 52.9147 ± 1.5359 | 51.5403 ± 2.9135 | 51.3744 ± 3.4172 |
| MIA (Test vs Retain) | 51.0896 ± 0.8636 | 52.0339 ± 2.3948 | 52.4334 ± 2.4510 | 53.0993 ± 1.6355 | 51.1622 ± 1.0741 |
| MIA (Train vs Test) | 53.0266 ± 0.0000 | 51.8523 ± 1.7339 | 53.1114 ± 1.2864 | 49.5521 ± 1.8352 | 52.2397 ± 1.9788 |
| **MIAU** | 0.1007 ± 0.0000 | 16.4583 ± 27.9277 | 24.9041 ± 32.1321 | 34.0881 ± 30.0245 | 19.1701 ± 18.4436 |

Table 26: Gradual unlearning on CIFAR10 ResNet-18 saliency

| Metric | Retrain 25% | Retrain 50% | Retrain 75% | Retrain |
|---|---|---|---|---|
| Retain Accuracy | 94.0417 ± 0.0432 | 94.0216 ± 0.0835 | 94.0349 ± 0.0973 | 94.0522 ± 0.0842 |
| Test Accuracy | 91.5527 ± 0.0000 | 91.5527 ± 0.0000 | 91.5527 ± 0.0000 | 91.5527 ± 0.0000 |
| Forget Accuracy | 93.9779 ± 0.7714 | 94.2178 ± 0.3940 | 94.1296 ± 0.4330 | 94.1055 ± 0.4248 |
| ZRF Score | 82.3755 ± 0.3442 | 82.4136 ± 0.2904 | 82.4678 ± 0.3234 | 82.4324 ± 0.3346 |
| | | | | |
| MIA (Forget vs Retain) | 72.5400 ± 1.4112 | 74.4500 ± 1.0742 | 74.2733 ± 0.7466 | 72.3050 ± 0.7672 |
| MIA (Forget vs Test) | 49.2200 ± 2.0558 | 49.9300 ± 1.1851 | 49.4467 ± 1.0390 | 49.7500 ± 0.6472 |
| MIA (Test vs Retain) | 66.1550 ± 0.6250 | 66.3100 ± 0.7190 | 66.2200 ± 0.7055 | 66.4275 ± 0.7870 |
| MIA (Train vs Test) | 51.0850 ± 0.1107 | 51.0725 ± 0.1017 | 51.1100 ± 0.1113 | 50.9700 ± 0.3295 |
| **MIAU** | 0.1049 ± 0.0949 | 11.7345 ± 15.5101 | 8.3344 ± 14.0206 | 40.0201 ± 21.0394 |

Table 27: Gradual unlearning on CIFAR20 AllCNN subclass

| Metric | Retrain 25% | Retrain 50% | Retrain 75% | Retrain |
|---|---|---|---|---|
| Retain Accuracy | 90.1986 ± 0.2034 | 90.2356 ± 0.1720 | 90.0492 ± 0.1297 | 90.0846 ± 0.1343 |
| Test Accuracy | 80.0192 ± 0.3384 | 80.0889 ± 0.1351 | 80.0922 ± 0.3936 | 79.9164 ± 0.2603 |
| Forget Accuracy | 86.5591 ± 3.7248 | 84.7336 ± 2.9078 | 83.9468 ± 1.2540 | 84.2852 ± 3.3768 |
| ZRF Score | 89.8222 ± 1.2087 | 89.6119 ± 0.2653 | 89.7607 ± 0.2431 | 89.5792 ± 0.5224 |
| | | | | |
| MIA (Forget vs Retain) | 65.3333 ± 10.0664 | 60.6667 ± 1.1547 | 57.8947 ± 1.3158 | 68.6667 ± 4.5092 |
| MIA (Forget vs Test) | 46.6667 ± 15.1438 | 52.0000 ± 8.7178 | 45.6140 ± 6.7521 | 53.0000 ± 2.0000 |
| MIA (Test vs Retain) | 49.8917 ± 0.5198 | 50.4750 ± 0.7233 | 50.7583 ± 0.9118 | 50.0917 ± 0.3166 |
| MIA (Train vs Test) | 54.7750 ± 0.5847 | 55.2000 ± 0.3500 | 54.7833 ± 0.4216 | 55.0417 ± 0.1283 |
| **MIAU** | 21.8945 ± 18.9314 | 11.6050 ± 19.6292 | 21.7195 ± 18.7950 | 77.7218 ± 19.2063 |

Table 28: Gradual unlearning on CIFAR20 AllCNN full class

| Metric | Retrain 25% | Retrain 50% | Retrain 75% | Retrain |
|---|---|---|---|---|
| Retain Accuracy | 90.1588 | 90.1953 | 95.2136 ± 6.7602 | 90.7568 |
| Test Accuracy | 79.8965 | 80.1951 | 75.8360 ± 4.6587 | 77.1298 |
| Forget Accuracy | 70.6888 | 64.0074 | 42.8687 ± 0.0911 | 0.0000 |
| ZRF Score | 91.9675 | 92.8224 | 93.5030 ± 0.9251 | 94.6184 |
| MIA (Forget vs Retain) | 51.2000 | 48.4000 | 73.0000 ± 21.5903 | 58.7000 |
| MIA (Forget vs Test) | 59.6000 | 59.4000 | 62.4667 ± 6.8825 | 68.6000 |
| MIA (Test vs Retain) | 51.8250 | 50.3750 | 62.7625 ± 19.4631 | 49.5750 |
| MIA (Train vs Test) | 55.2000 | 56.1500 | 65.7375 ± 15.8569 | 54.5250 |
| **MIAU** | 11.0710 | 33.2184 | 81.9300 | 99.8993 |