# OpenReview forum: "MIAU: Membership Inference Attack Unlearning Score for Quantifying the Forgetting Quality of Unlearning Methods"
_ICLR.cc/2026/Conference — Submitted to ICLR 2026_

### Official Review · Reviewer_YQ9P · 2025-10-27

**Soundness:** 2
**Presentation:** 3
**Contribution:** 2
**Rating:** 2
**Confidence:** 4

**Summary:**

The paper proposes MIAU (Membership Inference Attack Unlearning Score), a composite metric for evaluating machine unlearning. MIAU compares three MIA setups—Forget vs Test, Forget vs Retain, and Retain vs Test—then normalizes each attack’s accuracy between a baseline model trained on all data and a retrained model trained without the forget set. A logistic mapping turns a “gap-closure fraction” into a bounded 0–100 score, and the final MIAU is a weighted average (default equal weights). Experiments on MNIST, CIFAR-10/20, and MUCAC across ResNet-18, All-CNN, and ViT, plus several unlearning methods (Fine-tune, SSD, Amnesiac, Teacher), indicate MIAU can separate methods and is often monotone under “partial retraining” baselines.

**Strengths:**

- This paper introduces an interpretable metric that aggregates complementary MIA views and normalizes them between baseline and retrain references, yielding a single bounded score that’s easier to compare across methods and datasets.
- This paper proposes a practical audit$\rightarrow$deploy workflow: use a one-time retrained reference to select an unlearning method offline, then apply the chosen method in production—keeping evaluation principled while limiting operational overhead.

**Weaknesses:**

- The contribution feels largely constructive/standardizing rather than conceptually new: it consolidates existing MIA signals with anchoring and a bounded mapping to improve interpretability, which is valuable but incremental in novelty.
- The proposed audit$\rightarrow$deploy workflow reads as a sensible formalization of common practice in unlearning evaluation, offering operational clarity but not introducing a fundamentally new deployment paradigm.
- There is growing evidence that MIAs lose discriminative power as model capacity increases. The paper’s own ResNet vs. ViT results seem broadly consistent with this trend. It would strengthen the contribution to discuss the implications for MIAU at larger scales and, where feasible, include evaluations on substantially larger models or complementary privacy signals for regimes where MIA signal weakens.
- The placement of Figure 2 may be a bit early: it introduces notation before the symbols are formally defined (later in the text), which can make a first read harder to follow. Consider moving the figure to the methods section where definitions appear, or adding a brief notational reference in the caption so the figure is self-contained.
- For $\beta,\gamma,\delta$, the paper fixes equal weights but offers no sensitivity study or guidance to select weights for different privacy/utility trade-offs. This limits practical tunability.

[1] Duan, M., Suri, A., Mireshghallah, N., Min, S., Shi, W., Zettlemoyer, L., ... & Hajishirzi, H. (2024). Do membership inference attacks work on large language models?. arXiv preprint arXiv:2402.07841.

**Questions:**

See Weaknesses section.

**Details Of Ethics Concerns:**

NO or VERY MINOR ethics concerns only

---

### Official Review · Reviewer_rN33 · 2025-10-29

**Soundness:** 1
**Presentation:** 3
**Contribution:** 2
**Rating:** 2
**Confidence:** 4

**Summary:**

The paper introduces the Membership Inference Attack Unlearning Score (MIAU), a composite metric to assess machine unlearning quality. MIAU aggregates three pairwise MIA tasks: Forget vs. Test, Retain vs. Forget, and Retain vs. Test, and normalizes performance between a original model and a retrained model.
A logistic transformation maps gap closure into a 0–100 score. The authors propose MIAU as an offline auditing tool to select the best unlearning method for a model-dataset pair, avoiding repeated retraining in deployment. Experiments span image classification benchmarks (MNIST, CIFAR-10/20, MUCAC), model architectures (ResNet-18, All-CNN, ViT), and unlearning methods (Fine-tune, SSD, Amnesiac, Teacher). The paper claims MIAU captures gradual forgetting and overcomes limitations of single MIA evaluations.

**Strengths:**

- The paper clearly articulates failure modes of individual MIA tasks (Section 1.1) and motivates the need for a unified metric.
- The experimental scope is reasonable, covering multiple datasets, models, unlearning methods, and robustness checks with multiple iterations.
- Combining three complementary MIA tasks is a logical step beyond single-pair evaluations in prior work.

**Weaknesses:**

- The core contribution appears to be a calibration and aggregation of three existing MIA comparisons (Forget vs. Test, Retain vs. Forget, Retain vs. Test), which are already well-established in the literature, which cited in L58-L59. The normalization via gap closure and logistic transformation feels like a minor post-processing step rather than a substantive innovation.

- Although the paper asserts that MIAU detects imperfect unlearning where individual MIAs fail (e.g., retained representations despite low forget accuracy), but provides no targeted experiments to demonstrate this. For example, no adversarial unlearning setups, synthetic failure cases, or ablations are included. Without such evidence, the claimed improvement over raw MIAs remains unsubstantiated.

- In L325-L327, the paper expects MIAU to increase strictly with partial forgetting (MIAU_{25%} < MIAU_{50%} < MIAU_{75%}), but offers no theoretical justification. MIA accuracy reflects binary classification on differing logit distributions; there is no inherent reason for monotonic improvement. Diverse training dynamics in partial retraining may alter separability unpredictably. This assumption is critical to the “gradual forgetting” claim and requires formal derivation or counterexamples.

- Standard deviations frequently exceed 20 points on a 0–100 scale. This implies that the same unlearning method may receive scores differing by over 40 points across random seeds. Such instability prevents reliable method ranking and defeats the stated goal of efficient offline auditing.

**Questions:**

- Can you provide experiments where MIAU detects imperfect unlearning (e.g., retained internal representations with low forget accuracy) while individual MIAs fail?
- Why should MIAU increase monotonically with partial forgetting levels? Please include a theoretical derivation or counterexamples where this property fails.
- How can the high standard deviations in Table 1 be reduced for practical use?

---

### Official Review · Reviewer_XsVe · 2025-10-31

**Soundness:** 1
**Presentation:** 2
**Contribution:** 1
**Rating:** 2
**Confidence:** 4

**Summary:**

The paper proposes MIAU, an unlearning audit score that aggregates several MIAs and normalizes them between a baseline model and a fully retrained model, aiming to provide a quick offline way to compare unlearning methods without retraining.

**Strengths:**

1. The “offline audit --> choose a method --> deploy” workflow is well presented and could be convenient in practice.

2. The evaluation is broad, spanning multiple datasets and architectures, with partial-retrain references that acknowledge MIA limits and attempt graded validation.

**Weaknesses:**

1. The paper states that prior works “often rely on a single comparison and lack reference points (baseline/retrain).” However, this statement is inaccurate as many unlearning papers do compare to retrain/baseline, including (and not limited to):

[1] Fan, Chongyu, et al. "Salun: Empowering machine unlearning via gradient-based weight saliency in both image classification and generation." ICLR 2024.

[2] Zhao, Kairan, et al. "What makes unlearning hard and what to do about it." NeurIPS 2024.

2. The argument that Retain–Test is essential as a generalization sanity check is not so convincing when standard retain/test accuracies already capture this. Moreover, for the forget-retain setup, the paper suggested that effective unlearning should increase separability between forget and retain sets, but this also risks enabling attackers to identify the forget set, which undermines privacy goals. As a result, I don't think the justification for these two additional steups is strong enough.

3. There are some other questionable statements in the paper. E.g."after unlearning, accuracy on the forget set should drop slightly, ideally approaching test-level performance" This sounds vague and can be misleading, as matching forget-set performance to test-set performance is not generally a sound or universal unlearning target.

**Questions:**

1. You use a logit-based binary classifier as the MIA, but have you tried other (more advanced) MIAs? And how would the choice of MIA affect the results?

---

### Official Review · Reviewer_RBMY · 2025-10-31

**Soundness:** 3
**Presentation:** 2
**Contribution:** 2
**Rating:** 4
**Confidence:** 3

**Summary:**

This paper proposes MIAU (Membership Inference Attack Unlearning Score), a new metric designed to evaluate the effectiveness of machine unlearning methods. Unlike prior approaches that rely on a single MIA setting or raw accuracy differences, MIAU integrates three complementary MIA comparisons, Forget vs Test, Forget vs Retain, and Retain vs Test, to capture residual memorization, removal specificity, and generalization stability. The key idea is to position an unlearned model between two meaningful reference points: the baseline model trained on all data and the fully retrained model without the forget set. Using a “gap‐closure fraction” and a calibrated logistic transformation, MIAU provides a normalized, interpretable score (0–100) indicating how closely an unlearning method approximates the privacy behavior of full retraining. Extensive experiments across datasets, architectures, and unlearning methods demonstrate that MIAU can differentiate strong and weak unlearning approaches and generally increases under partial retraining, reflecting progressive forgetting.

**Strengths:**

(1)One of the most valuable aspects of the proposed MIAU score is the normalization between the fully trained model and the fully retrained model. This creates a principled evaluation interval and enables intuitive interpretation of “how much forgetting has been achieved.”

(2)The authors evaluate MIAU on multiple benchmarks, including MNIST, CIFAR-10/20, and MUCAC, and across diverse model families such as ResNet, AllCNN, and ViT. This broad coverage enhances the external validity of the findings.

**Weaknesses:**

(1)The proposed MIAU score entirely depends on membership inference attacks, inheriting several well-known limitations of MIAs themselves, such as instability across random seeds, sensitivity to data complexity, calibration issues, and weak discriminative power on well-generalized models (e.g., MNIST). As a result, the reliability of MIAU is fundamentally constrained by the weaknesses of its underlying signal.

(2)The method implicitly assumes that a retrained model fully represents “ideal forgetting,” but this assumption does not necessarily hold. Retraining may still capture distribution-level information about the forgotten data or produce nontrivial MIA signals. Thus, anchoring the metric strictly between baseline and retrain introduces a conceptual bias and may misrepresent unlearning effectiveness in certain settings.

(3)Different applications impose different priorities: privacy-focused scenarios emphasize Forget vs Test, while utility-centric ones emphasize Retain vs Test. Using fixed uniform weights lacks methodological grounding and may mask important trade-offs.

**Questions:**

The experimental setup largely relies on weak or limited MIAs, uses small and simple datasets, includes only a narrow set of unlearning baselines, and lacks evaluation on more recent or complex models.

---

### Meta-Review · Area_Chair_H9Q1 · 2025-12-27

**Summary:**

There is broad reviewer consensus that the contribution is incremental. Reviewers also raise some major concerns about soundness and reliability, including large variance across random seeds, lack of theoretical justification, and missing targeted experiments demonstrating that MIAU captures unlearning failures where individual MIAs do not. While useful as a consolidation of existing MIA-based evaluations, the work does not meet the acceptance bar for ICLR.

No author rebuttal was submitted to address these concerns.

**Reviewer Concerns:**

No author rebuttal was submitted to address these concerns.

**Reviewer Scores:**

No author rebuttal was submitted to address these concerns.

---

### Decision · Program_Chairs · 2026-01-26

Reject